# Residue proximity information and protein model discrimination using saturation-suppressor mutagenesis

Anusmita Sahoo[1], Shruti Khare[1], Sivasankar Devanarayanan[1], Pankaj C. Jain[1], Raghavan Varadarajan[1,2]*

[1] Molecular Biophysics Unit, Indian Institute of Science, Bangalore, India; [2] Jawaharlal Nehru Center for Advanced Scientific Research, Bangalore, India

**Abstract** Identification of residue-residue contacts from primary sequence can be used to guide protein structure prediction. Using *Escherichia coli* CcdB as the test case, we describe an experimental method termed saturation-suppressor mutagenesis to acquire residue contact information. In this methodology, for each of five inactive CcdB mutants, exhaustive screens for suppressors were performed. Proximal suppressors were accurately discriminated from distal suppressors based on their phenotypes when present as single mutants. Experimentally identified putative proximal pairs formed spatial constraints to recover >98% of native-like models of CcdB from a decoy dataset. Suppressor methodology was also applied to the integral membrane protein, diacylglycerol kinase A where the structures determined by X-ray crystallography and NMR were significantly different. Suppressor as well as sequence co-variation data clearly point to the X-ray structure being the functional one adopted *in vivo*. The methodology is applicable to any macromolecular system for which a convenient phenotypic assay exists.

*For correspondence: varadar@ mbu.iisc.ernet.in

## Introduction

Deducing the native conformation of a protein provides insight into its biological function. X-ray crystallography, NMR and more recently Cryo-EM (*Bartesaghi et al., 2015*; *Cheng, 2015*) are the major techniques used to accomplish this at atomic resolution. However, these require soluble purified protein at high concentration. In a few cases (*Garbuzynskiy et al., 2005*), the structures solved by different methods do not agree with each other, for various reasons. In many cases, the folded conformation is believed to be at a global free energy minimum (*Anfinsen, 1973*). This can be used as a guide to either deduce the structure (*Havel et al., 1979*) or fold a protein *in silico* (*Das et al., 2009*; *Das and Baker, 2008*; *Pande et al., 2003*).

The structural and functional integrity of a protein requires maintenance of specific interactions during the course of evolution. Evolution allows either conservation of these interacting pairs of residues or mutation at interacting positions in a correlated manner (*Godzik and Sander, 1989*; *Melero et al., 2014*). The fitness cost of most amino acid substitutions depends on the genetic context in which they occur. Substitutions beneficial in one background can be detrimental in a different background (*Breen et al., 2012*; *Shah et al., 2015*). Interestingly, some sites are conserved not because a given amino acid is functionally irreplaceable, but rather because the right context is not available for its evolution (*Wellner et al., 2013*). This provides further evidence of correlation amongst mutations, that is epistatic interactions.

Often, the detrimental effects of a mutation can be alleviated by a second (suppressor) mutation which compensates for defects in stability, packing or function caused by the first mutation. Such correlated substitutions have been experimentally identified in attempts to screen for second-site

**eLife digest** Common techniques to determine the three-dimensional structures of proteins can help researchers to understand these molecules' activities, but are often time-consuming and do not work for all proteins. Proteins are made of chains of amino acids. When a protein chain folds, some of these amino acids interact with other amino acids and these contacts dictate the overall shape of the protein. This means that identifying the pairs of contacting amino acids could make it possible to predict the protein's structure.

Interactions between pairs of contacting amino acids tend to remain conserved throughout evolution, and if a mutation alters one of the amino acids in a pair then a 'compensatory' change often occurs to alter the second amino acid as well. Compensatory mutations can suggest that two amino acids are close to each other in the three-dimensional shape of a protein, but the computational methods used to identify such amino acid pairs can sometimes be inaccurate.

In 2012, researchers generated mutants of a bacterial protein called CcdB with changes to single amino acids that caused the protein to fail to fold correctly. Now, Sahoo et al. – who include two of the researchers involved in the 2012 work – have developed an experimental method to identify contacting amino acids and use the CcdB protein as a test case. The approach involved searching for additional mutations that could restore the activity of five of the original mutant proteins when the proteins were produced in yeast cells. The rationale was that any secondary mutations that restored the activity must have corrected the folding defect caused by the original mutation. Sahoo et al. then predicted how close the amino acids affected by the secondary mutations were to the amino acids altered by the original mutations. This information was used to select reliable three-dimensional models of CcdB from a large set of possible structures that had been generated previously using computer models.

Next, the technique was applied to a protein called diacylglycerol kinase A. The structure of this protein had previously been inferred using techniques such as X-ray crystallography and nuclear magnetic resonance, but there was a mismatch between the two methods. Sahoo et al. found that the amino acid contacts derived from their experimental method matched those found in the crystal structure, suggesting that the functional protein structure in living cells is similar to the crystal structure. In the future, the experimental approach developed in this work could be combined with existing methods to reliably guide protein structure prediction.

suppressors (*Araya et al., 2012*; *Hecht and Sauer, 1985*; *Machingo et al., 2001*; *Pakula and Sauer, 1989*; *Sideraki et al., 2001*). Correlated mutations have also been employed to identify protein interaction sites (*Melamed et al., 2015*). Two other recent studies have described a large number of second-site suppressors for the RRM2 domain of the yeast poly(A)-binding protein (Pab1) and the IgG-binding domain of protein G (GB1), respectively (*Melamed et al., 2013*; *Olson et al., 2014*). While there have been past efforts to identify compensatory mutations, a comprehensive method to explicitly identify suppressors and to distinguish suppressors that are spatially proximal from those that are distal to the site of the original inactive mutation is lacking. In the current work, we attempt to address these issues.

In addition to experimental studies to identify correlated mutations, several computational methods also exist which detect coevolution of residues in homologous sequences. Various statistical approaches which identify correlated mutations have been used to decode such evolutionary information contained in multiple homologous sequences of a protein (*Göbel et al., 1994*). Progressive improvement in this area has resulted in more accurate strategies such as use of a Bayesian network model (*Burger et al., 2010*), maximum entropy in DCA (*Morcos et al., 2011*) and EVfold (*Marks et al., 2011*), sparse inverse covariance estimation in PSICOV (*Jones et al., 2012*) and a pseudo-likelihood based approach in GREMLIN (*Kamisetty et al., 2013*) to predict residue-residue contacts to computationally build protein 3D structures (*Marks et al., 2011*; *Jones et al., 2012*; *Ovchinnikov et al., 2014*; *Sulkowska, et al., 2012*). A sparse network of co-evolving residues of a protein constraining its structure, specificity and function has been examined by statistical coupling analysis of evolutionarily rich sequence data in protein families (*Halabi et al., 2009*). Although these

methods show great promise in the area of macromolecular structure determination, the fidelity of the predictions is questionable for candidates with small protein families, where the total number of sequences is less than five times the length of the protein in case of GREMLIN (*Kamisetty et al., 2013*) or less than 1000 for some of the other methods (*Morcos et al., 2011*). Further, stringent constraints present during natural evolution limit the number and nature of sequences that are sampled. We propose alternative experimental methodology termed saturation suppressor mutagenesis to complement and extend residue contact information obtained from computational analyses of correlated mutations in a multiple sequence alignment.

The test protein in this study, Controller of Cell Division or Death B (CcdB) is the toxin component of the *Escherichia coli* CcdA-CcdB antitoxin-toxin system. It is a globular, dimeric protein with 101 residues per protomer, involved in maintenance of F plasmid in cells by a mechanism involving its binding to and poisoning of DNA Gyrase (*Dao-Thi et al., 2005*). CcdB has two primary ligands with overlapping binding sites. These are its cognate antitoxin, CcdA and its cellular target, DNA Gyrase. CcdB has ~350 homologs in its protein family, which is less than five times the length of the protein. Therefore, computational methods cannot be reliably employed to deduce correlated mutations from a multiple sequence alignment of CcdB homologs.

Using CcdB, we identify spatially proximal residues by doing an exhaustive search for suppressor mutations for a given inactive mutant (*Figure 1*). We subsequently use this information for protein model discrimination and structure prediction. We have previously constructed a saturation mutagenesis library of CcdB in which each residue was individually randomized. The library was screened at multiple expression levels in *E. coli* and relative activities of ~1430 individual mutants were inferred by deep sequencing (*Adkar et al., 2012*). It was observed that the mutational sensitivity of each residue of CcdB is related to its depth from the protein surface. Residue depth is defined as the distance of any atom/residue to the closest bulk water (*Chakravarty and Varadarajan, 1999*; *Tan et al., 2011*). An empirical parameter (RankScore) related to mutational sensitivity was defined for each residue, which could be reliably used as a proxy for the residue's depth, provided the residue was not part of the active-site. Active-site residues could be deduced solely from mutational sensitivity data. A brief description of the screening procedure and definitions of mutational sensitivity score and RankScore are provided in the Materials and methods section.

In the present study, residues likely to be present at the core of CcdB were identified by their mutational sensitivity in the single mutant library (*Adkar et al., 2012*). Inactive mutants at five such putative buried residues were identified. Next, exhaustive second-site saturation mutagenesis libraries constructed in the background of each inactive mutant were displayed on the surface of yeast (*Chao et al., 2006*) and screened for binding to the CcdB ligand, DNA Gyrase, to identify suppressors. Distal and proximal suppressors could readily be distinguished based on the RankScore of the suppressor site in the original single-site saturation mutagenesis library (*Adkar et al., 2012*). The residue proximity information thus obtained, was used to generate spatial constraints, which were in turn able to capture the protein's native structural features and recovered >98% of native-like models (with backbone RMSD within 2.5 Å of the native structure) from a pool of decoys. Thus, high-throughput, exhaustive application of this approach, combined with ways to disentangle proximal from distal suppressors would be a useful way to identify a substantial number of distance constraints, which can be used for protein structure prediction (*Marks et al., 2011*).

Diacylglycerol kinase A (DgkA) is a homotrimeric integral transmembrane protein (121 residues per protomer) in *E. coli*, catalyzing the phosphorylation of diacylglycerol to phosphatidic acid. Gram-negative bacteria use this reaction product to shuttle water-soluble components to membrane-derived oligosaccharide and lipopolysaccharide in their cell envelope (*Van Horn and Sanders, 2012*). The protein has been captured in two distinct conformations by X-ray crystallography (PDB id 3ZE5 [*Li et al., 2013*]) and NMR (PDB id 2KDC [*Van Horn et al., 2009*]), respectively. The two structures are significantly different from each other, with 'domain swapping' being the key feature of the NMR model. Each structure is characterized by several unique residue contacts. It is important to identify whether one or both these structures are present in-vivo. We therefore isolated inactive mutants at residues involved in differential contacts in the two structures. Next, we experimentally identified suppressors of these inactive mutants. The residue contact pairs identified via these suppressors were consistent only with the contacts found in the crystal structure, thereby suggesting that the X-ray structure represents the native, active conformer of this membrane protein in-vivo.

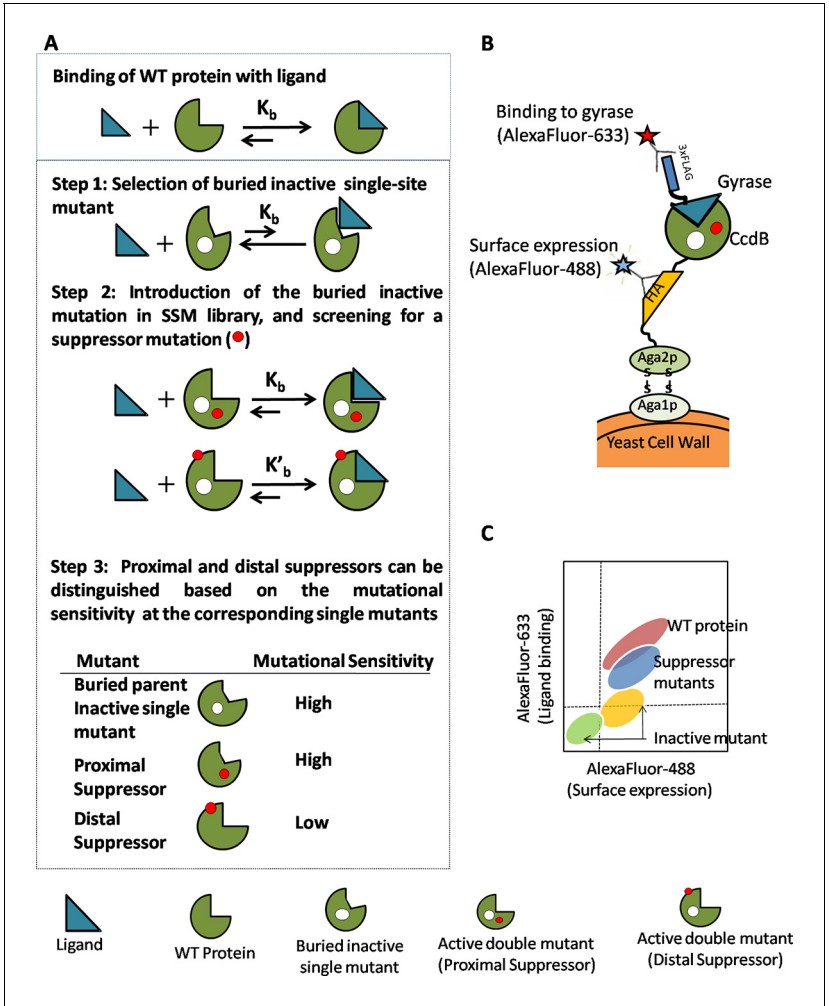

**Figure 1.** Strategy adopted to determine proximal residue pairs. (**A**) Parent inactive mutant binds cognate ligand with lower apparent affinity than Wild-Type protein (WT). (Parent inactive mutant, Suppressor mutant) pair binds ligand with higher affinity than the parent inactive mutant. Refer to *Figure 1—figure supplement 3* for the PCR strategy to introduce a parent inactive mutant into the single-site saturation mutagenesis (SSM) library of CcdB. Proximal and distal suppressors can be discriminated based on the mutational sensitivity of the corresponding single mutants. (**B**) Yeast Surface Display is used as the screening system in case of CcdB. (**C**) Suppressor mutants are identified using Fluorescence Activated Cell Sorting (FACS) of the mutant library using yeast surface display. Active-site residues can be distinguished from buried ones based on the pattern of mutational sensitivity (MS$_{seq}$). Putative active-site, buried and exposed residue positions were identified from the MS$_{seq}$ data. Representative plots are shown in *Figure 1—figure supplement 1*. Possible mechanisms responsible for reduced activity of a mutant protein are described in *Figure 1—figure supplement 2*.

The following figure supplements are available for figure 1:

**Figure supplement 1.** Barplots depicting mutational sensitivity (MS$_{seq}$) values of all available mutants at representative positions of CcdB.

**Figure supplement 2.** Possible mechanisms for reduced activity of a mutant protein.

**Figure supplement 3.** Strategy to introduce an inactive mutant into all members of a Single-site Saturation Mutagenesis (SSM) library of CcdB cloned in a yeast surface display (YSD) vector (pPNLS).

# Results

## Selection of parent inactive mutant

In the current work, we aimed at identifying suppressors for buried inactive non-active site residues of CcdB. Since WT CcdB is toxic to *E. coli* only cells transformed with inactive mutants will survive, thus providing a facile selection for such mutants. Relative activities of ~1430 single-site mutants of

CcdB have been obtained previously from phenotypic screening of a single-site saturation mutagenesis library at multiple expression levels in *E. coli*. Deep sequencing was used to determine the identities of all surviving (inactive) mutants at each expression level (*Adkar et al., 2012*). An empirical parameter (RankScore) was defined for each residue (see Materials and methods). It is related to the average mutational sensitivity (see Materials and methods) at each position. It was observed that for non-active site positions, RankScore is proportional to residue depth.

The sequencing data from the above library was analyzed to select five inactive mutants; V5F, V18W, V20F, L36A and L83S (hereafter referred to as parent inactive mutants) at buried, non-active site residues. Several buried, non-active site residues were identified from sequencing data. Parent inactive mutants were chosen so as to include different kinds of mutations, namely large→small, small→large and hydrophobic→polar. Patterns of mutational sensitivity ($MS_{seq}$) for mutants at a position were analysed to distinguish between active-site and buried mutations as described below.

Both buried and active-site residue positions possess high RankScores and high average mutational sensitivity ($MS_{seq}$) values. Active-site residues can be distinguished from buried ones based on the pattern of mutational sensitivity. Mutational sensitivity ($MS_{seq}$) scores were determined from the sequencing analysis of the single-site saturation library of CcdB and refer to the expression level at which partial loss of function mutants show an active phenotype (methodology described in [*Adkar et al., 2012*]). Patterns of $MS_{seq}$ values for active-site, buried and exposed positions differ from each other. Representative plots are shown in *Figure 1—figure supplement 1*. At buried positions, typically most aliphatic substitutions are tolerated; hence they have low $MS_{seq}$ values. Polar and charged residues are poorly tolerated at buried positions, thus showing high $MS_{seq}$ values (*Figure 1—figure supplement 1A*). By contrast, mutations to aliphatic residues are often poorly tolerated at active-site residues (which are typically exposed) and have higher $MS_{seq}$ values (*Figure 1—figure supplement 1B*). Polar and charged residues are sometimes tolerated and the average mutational sensitivity for active-site residues is typically higher than for buried residues. By analyzing the mutational sensitivity patterns at all positions, Q2, F3, Y6, S22, I24, N95, W99, G100 and I101 can be identified as putative active-site residues. These residues were identified based solely on mutational data (unpublished results). Similar mutational patterns are seen for at least two other proteins for which extensive mutational data exist, the PDZ domain (PSD95$^{pdz3}$) (*McLaughlin et al., 2012*) and the IgG-binding domain of protein G (GB1) (*Olson et al., 2014*) (unpublished results).

Recent work (*Melamed et al., 2015*) suggests that saturation mutagenesis in combination with evolutionary conservation data can also be used to identify residues at interaction sites. In addition to differences in mutational sensitivity patterns which are employed here to discriminate between active-site and buried residues, an important difference between active-site and buried mutations is that the former typically affect specific activity and not the level of properly folded protein, while the latter primarily affect the level of properly folded protein (*Bajaj et al., 2008*). Thus, measurements of protein levels, and possibly sensitivity of mutant activity to chaperone overexpression (*Tokuriki and Tawfik, 2009*), can also be used to distinguish between active-site and buried mutants. The average hydrophobicity and hydrophobic moment (*Varadarajan et al., 1996*) are supplementary parameters that can help distinguish between exposed, active-site and buried residues.

## Library preparation and isolation of suppressor mutants of CcdB

Mutations at the selected non-active site positions perturb activity by reducing the amount of functional, folded protein *in vivo* (*Figure 1—figure supplement 2*) (*Bajaj et al., 2008*). In order to identify residues which can compensate for the parent inactive mutant, the selected parent inactive mutations described above were individually introduced into the single-site saturation mutagenesis library constructed previously (*Adkar et al., 2012*). The resulting double mutant saturation-suppressor libraries were cloned into a yeast surface display vector (pPNLS) by three fragment recombination in *Saccharomyces cerevisiae* (*Figure 1—figure supplement 3*). Recombination yielded ~$10^5$ transformants for each library. Yeast surface display (*Chao et al., 2006*) and Fluorescence Activated Cell Sorting (FACS) were used as screening tools to isolate populations exhibiting (i) enhanced binding to the ligand DNA Gyrase, and (ii) increased surface expression relative to the corresponding parent inactive mutant. The ligand concentration was decreased in subsequent rounds of sorting to increase the stringency in selection of true suppressors (*Supplementary file 1*). The methodology used to generate the library along with the use of very low (pg) amount of template DNA for PCR was effective in introducing the parent inactive mutant into all members of the single-site saturation

mutagenesis library. This is important as the presence of even a small fraction of WT residue at the position of the parent inactive mutant will rapidly lead to its selection, resulting in a high amount of false positive data. Following multiple rounds of sorting (typically 3–4), over 10% of the population was shown to bind significantly better to DNA Gyrase than the parent inactive mutants. Data for the L83S mutant library is shown (*Figure 2*). At this stage, 96 individual clones for each library were sequenced by Sanger sequencing to identify 1–3 potential suppressors for each parent inactive mutant (*Table 1*).

## Discrimination between proximal and distal suppressors

For a pair of residues in contact, it is likely that a destabilizing substitution at one residue can be suppressed by a substitution with complementary properties at the other residue. For example, a cavity formed by a large→small substitution of a residue may be compensated by a small→large substitution of the partner residue in contact with it. Consequently, while each of the individual mutations will be destabilizing in the WT background, the pair will have increased stability and activity relative to their corresponding single mutants, thus demonstrating positive sign epistasis.

However, suppressors can be either spatially proximal or distal from the site of the original inactive mutation. In contrast to proximal suppressors, distal suppressors will typically be on the surface of protein (*Bank et al., 2015*) and, hence, the individual suppressor mutation is expected to be neutral in the WT background. Further, unlike proximal suppressors no complementarity in amino acid property relative to the parent inactive mutant is expected for a distal suppressor. In the present study, we use RankScore values to discriminate between proximal and distal suppressors. The approach is validated by mapping proximal and distal suppressors onto the known crystal structure of CcdB.

We have previously shown that the RankScore parameter correlates with residue depths derived from the crystal structure of the protein (Figure 4B of [*Adkar et al., 2012*]). The value for RankScore

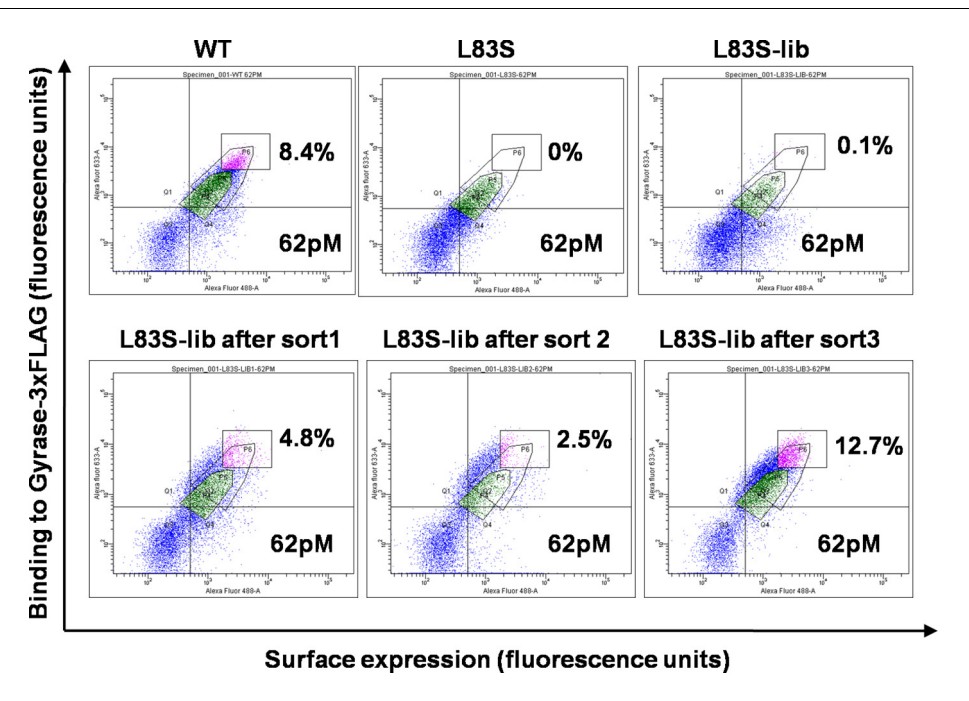

**Figure 2.** Enrichment of second-site suppressor population for L83S CcdB inactive mutant following multiple rounds of Fluorescence Activated Cell Sorting (FACS). 62 pM of Gyrase-3xFLAG was used in all cases for analysis. 4 nM, 1 nM and 0.25 nM Gyrase-3xFLAG was used for the three rounds of sort respectively. The population gated in green shows the parent inactive mutant L83S, while that in magenta shows the sorted population which has higher surface expression and tighter binding to the ligand than L83S. Sort details for different libraries of CcdB are summarized in *Supplementary file 1*.

**Table 1.** Experimentally determined (Parent inactive mutant, suppressor) pairs for CcdB occur at both spatially proximal and distal residues.

| Parent inactive mutant (X) | Suppressor Mutant (Y) | ASA (%) | Depth (Å) | RankScore[a] | Shortest distance[b] (Å) | Centroid-centroid distance[c] (Å) |
|---|---|---|---|---|---|---|
| V5F[d] | | 0 | 6.8 | 54 | | |
| | L36M[d,e] | 0 | 7.3 | 83 | 3.9 | 5.5 |
| | A81G[d,e] | 32.4 | 3.9 | 26 | 2.8 | 4.2 |
| V18W | | 0 | 9.3 | 59 | | |
| | M63T[e] | 0.1 | 8.1 | 46 | 4.2 | 5.8 |
| | I90V[e] | 0.1 | 7.4 | 60 | 4.8 | 5.9 |
| V20F | | 0 | 8.6 | 60 | | |
| | E11R[f] | 112.7 | 3.4 | 1 | 18.4 | 22.4 |
| | E11K[f] | 112.7 | 3.4 | 1 | 18.4 | 22.4 |
| L36A | | 0 | 7.3 | 83 | | |
| | M63L[e] | 0.1 | 8.1 | 46 | 3.8 | 6.4 |
| | R10G[f] | 76.6 | 3.6 | 1 | 11.6 | 18.1 |
| | E11P[f] | 112.7 | 3.4 | 1 | 11.7 | 16.8 |
| L83S | | 1.5 | 5.8 | 42 | | |
| | V54L[e] | 0.4 | 5.7 | 28 | 3.9 | 4.5 |

[a]RankScore for residues X or Y estimated from phenotypic screening of single-site saturation mutagenesis library of CcdB and deep sequencing (**Adkar et al., 2012**)

[b]Shortest distance between residues X and Y

[c]Distance between side chain centroids of residues X and Y

[d]The suppressors (Y) were identified as a triple mutant with the parent inactive mutant (X) V5F, that is V5F/L36M/A81G

[e]Suppressor residues spatially proximal to parent inactive mutant

[f]Suppressor residues distal from the parent inactive mutant

Heavy atoms of the residues are considered for calculating distances using the crystal structure of CcdB, PDB id 3VUB (**Loris et al., 1999**).

ranges between 1 to 100. For a given non-active site residue, a higher value of RankScore indicates that the residue is likely to be buried in the protein structure. All the parent inactive mutants were chosen such that they are non-active site and have a high RankScore. These residues are predicted to be buried and indeed are found to be buried in the crystal structure. It is therefore expected that positions at which local suppressors occur will also have high depth, high RankScores and high average mutational sensitivity in the single-site saturation mutagenesis library (**Figure 1A**).

All the positions with low RankScores are found to be exposed on the protein structure. Hence, suppressors with RankScore=1 were classified as distal suppressors (**Table 1**). A RankScore of 1 is a conservative cutoff for distal suppressors, probably a cutoff of five or ten would yield similar results. The RankScore cutoff of 25 which we have chosen for proximal suppressors, clearly identifies only buried residues with depth > 5.5 Å. All these residues have accessibility < 1.5%.

The PCR strategy adopted (**Figure 1—figure supplement 3**) to construct the second-site suppressor mutagenesis library for screening suppressors against the parent inactive mutants of CcdB was designed to generate (i) 50% double mutants with each member containing the parent inactive mutant and a single mutant, and (ii) 50% parent inactive mutants. Sequencing data obtained was analyzed to identify proximal and distal suppressors (as discussed above). We largely obtained double mutants containing the parent inactive mutant and a suppressor, and a small fraction of the triple mutant (V5F/L36M/A81G) (**Table 1**). Occurrence of the triple mutant was due to selection and enrichment of an additional mutation introduced into the double mutant library likely due to PCR based errors. The selected triple mutant contained the parent inactive mutant V5F and two proximal suppressors (L36M and A81G). Overall, six residue pairs were identified, which contained the parent inactive mutant and a putative proximal suppressor (**Table 1**).

The shortest distance between individual members of the identified (parent inactive mutant, proximal suppressor) pairs in the structure of WT CcdB was 2.8–4.8 Å (PDB id: 3VUB (*Loris et al., 1999*), *Table 1*, *Figure 3*). This validated the methodology described above. Two residues, R10 and E11 (in the pairs L36A/R10G, L36A/E11P, V20F/E11R and V20F/E11K) were identified as locations for distal

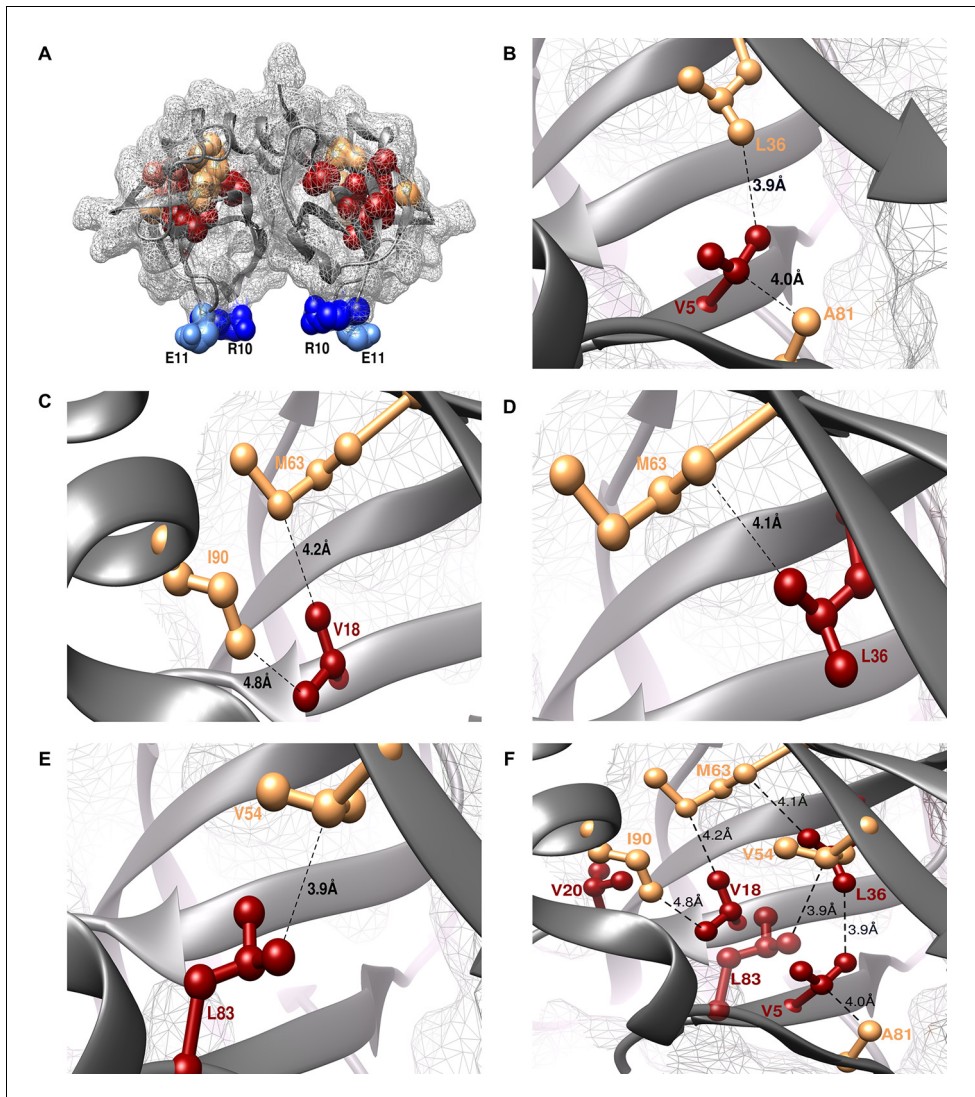

**Figure 3.** Experimentally obtained (Parent inactive mutant, suppressor) pairs mapped onto the crystal structure of CcdB (PDB id 3VUB [*Loris et al., 1999*]). Only the Wild-Type (WT) residues of each member of the pair are shown as structures of the mutants are not available. (**A**) The two monomers of the dimeric CcdB protein are shown in ribbon (dark grey) with the distal suppressors (R10 (dark blue) and E11 (light blue) mapped on an exposed loop region while the proximal suppressors (light brown) and parent inactive mutants (dark red) are present in the core of the protein. One of the monomers of the dimeric CcdB protein has been shown in B-F for ease of visualization of the mapped parent inactive mutants and corresponding proximal suppressors. (**B**) Parent inactive mutant V5 and its proximal suppressors L36 and A81, (**C**) Parent inactive mutant V18 and its proximal suppressors M63 and I90, (**D**) Parent inactive mutant L36 and its proximal suppressor M63, (**E**) Parent inactive mutant L83 and its proximal suppressor V54 are shown. Dotted lines indicate the shortest distance between the two residues in each pair. (**F**) (Parent inactive mutant, proximal suppressor) pairs are clustered in the core of the protein forming an interconnected network, with side chains facing towards each other. Residues common to multiple (Parent inactive mutant, suppressor) pairs, for example L36 and M63, provide additional information about the network of interactions. The figure has been prepared using the UCSF Chimera package (developed by Resource for Biocomputing, Visualization, and Informatics at the University of California, San Francisco [*Pettersen et al., 2004*]).

suppressors based on their low RankScore values in the single-site saturation mutagenesis library. Mutations at E11 were observed to suppress parent inactive mutants at multiple residue positions. The shortest distances between L36-R10 and V20-E11 are 11.6 Å and 18.4 Å, respectively. As expected, the proximal suppressors are clustered together in the protein interior while the distal ones are present on an exposed loop region (*Figure 3A*, *Video 1*).

The ability to suppress parent inactive mutants at multiple positions is indicative of the suppressor being a global suppressor. The most reliable way to identify these would be to confirm that the same putative distal suppressor is able to suppress multiple parent inactive mutants, preferably the ones which are not in contact with each other. We treated both the distal suppressors (R10 and E11) as likely global suppressors and performed experiments described in the following section to test this hypothesis.

## Protein stabilization by the suppressor mutants

Identified CcdB suppressors individually conferred improved binding and thermal stability to the corresponding parent inactive mutants (*Figure 4*, *Figure 4—figure supplement 1*, *Figure 4—figure supplement 2*, *Supplementary file 2*). An increased affinity for DNA Gyrase for the (Parent inactive mutant, suppressor) pair relative to the parent inactive mutant alone was observed in all cases except for (L36A, M63L) and (L36A, R10G) pairs (*Figure 4C*), which exhibited similar affinity towards Gyrase as the parent inactive mutant L36A. However, the (Parent inactive mutant, suppressor) pairs showed higher thermal stability ($T_m$(L36A, M63L)=55 ± 0.8°C, $T_m$(L36A, R10G)=55.1 ± 0.4°C, $T_m$(L36A)=47.1 ± 0.3°C) and increased surface expression (*Figure 4A,D*). Surface expression levels of proteins displayed on the yeast surface have previously been found to correlate with the protein's stability (*Shusta et al., 1999*). Increased surface expression was observed for all (Parent inactive mutant, suppressor) pairs (including the distal suppressor (L36A, R10G) pair) relative to the parent inactive mutant, except for the (V20F, E11R) pair (*Figure 4A,B*). This distal suppressor pair exhibited slightly lower expression than its parent inactive mutant, V20F, but displayed enhanced activity in terms of its binding to Gyrase (*Figure 4B,C*).

R10G is a distal suppressor. The ability of the R10G mutation to suppress defects at other positions was examined by constructing the corresponding double mutants. Increased surface expression of R10G paired with each of V5F, V18W, L36A and L83S was seen relative to the individual parent inactive mutants (*Figure 4A,B*). This demonstrates that R10G likely acts as a global suppressor. E11 suppresses activity of two parent inactive mutants, L36A and V20F (L36A/E11P, V20F/E11R, V20F/E11K) and is anticipated to play a role similar to R10G.

## Characterization of distal suppressors

The presence of the CcdB ligand, CcdA during thermal denaturation of CcdB, shifts the unfolding equilibrium towards the folded fraction of CcdB, resulting in an increased $T_m$ than when monitored in its absence (*Supplementary file 2*, *Figure 4—figure supplement 2*). This increase in $T_m$ was observed for all mutants except R10G. L36A/R10G showed an increase of only 9°C in the presence of CcdA while other mutants showed an increase of >20°C. These observations indicated a decrease in affinity of R10G CcdB for CcdA. The distal suppressors R10G and E11R are present on an exposed loop of CcdB and contact CcdA in the crystal structure of the CcdB-CcdA complex, PDB id 3G7Z (*De Jonge et al., 2009*). R10 forms a hydrogen bond (2.9 Å) with N69 of CcdA. Hence, mutations at R10 and E11 are likely to destabilize binding of CcdB to CcdA, consistent with our results. The suppressor mutant, R10G has an increased stability relative to WT (R10G ($T_m$(R10G)=74.8 ± 0.2°C, $T_m$(WT)=66.8 ± 1.0°C, *Supplementary file 2*, *Figure 4D*). However, the

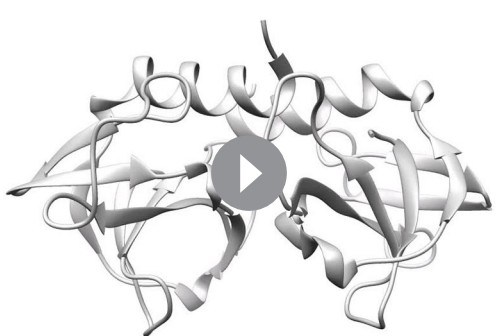

**Video 1.** Experimentally obtained (Parent inactive mutant, suppressor) pairs mapped onto the crystal structure of CcdB (PDB id 3VUB), related to *Figure 3*. Parent inactive mutant is abbreviated as PIM in the video.

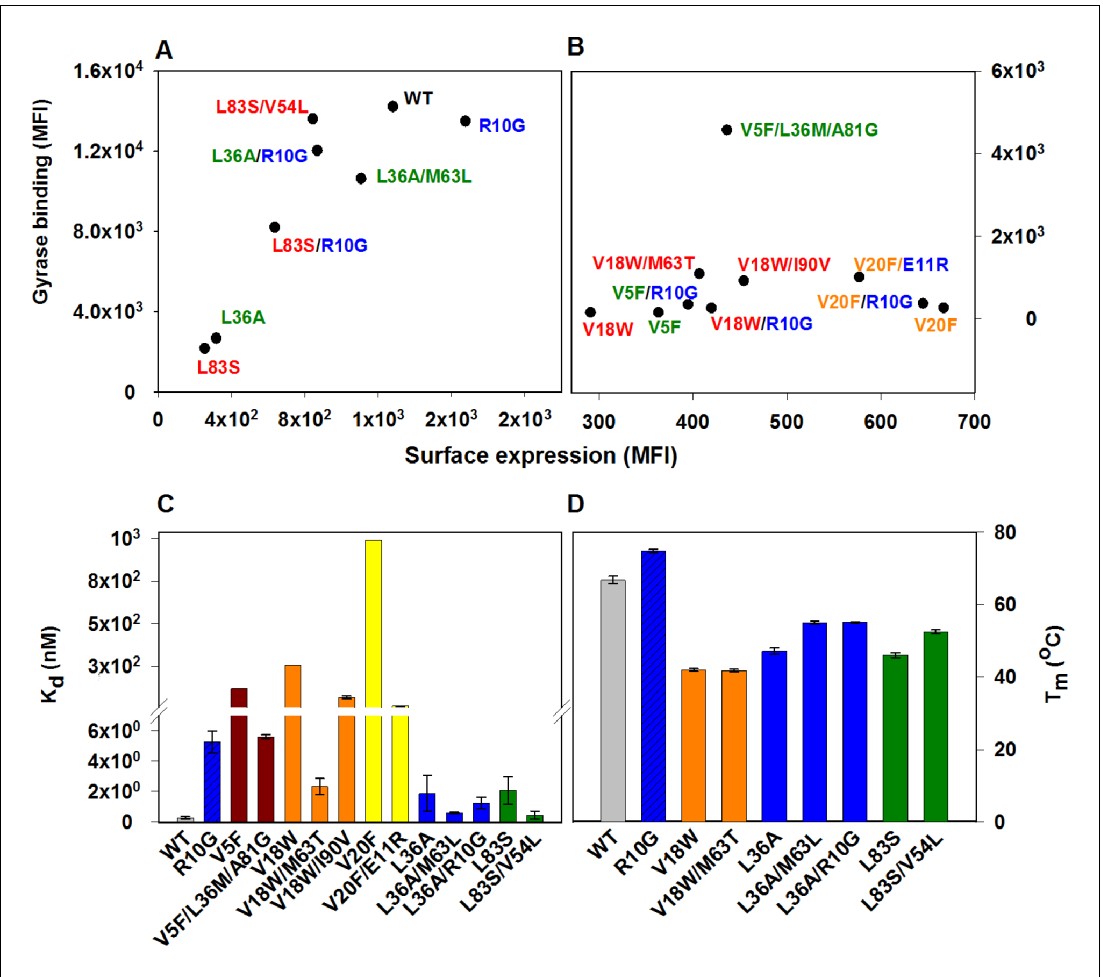

**Figure 4.** Restoration of defects in CcdB parent inactive mutants by suppressors. Proteins displayed on the yeast surface were monitored by their surface expression (abscissa) and Gyrase binding (ordinate) signals in terms of Mean Fluorescence Intensity (MFI) by Fluorescence Activated Cell Sorting (FACS). In order to express the mutant of interest prior to sorting/analysis, yeast cells were induced at (**A**) 30°C for more stable parent inactive mutants and their suppressor pairs or (**B**) at 20°C for less stable parent inactive mutants and corresponding suppressor pairs. Suppressor mutations result in enhanced ligand binding and surface expression. In each (Parent inactive mutant, suppressor) pair, parent inactive mutants and corresponding proximal suppressors are denoted with the same color. Distal suppressors R10G and E11R are colored in blue in (**A**) and (**B**). MFI values averaged over two independent experiments are shown. Suppressor mutations result in increased (**C**) ligand affinity and (**D**) thermal stability. (**C**) $K_d$ for Gyrase measured using yeast surface display titrations. Refer to *Figure 4—figure supplement 1* for yeast surface display titrations of parent inactive mutants and (Parent inactive mutant, suppressor) pairs. (**D**) Thermal melting temperatures ($T_m$) for parent inactive mutants in the absence and presence of second-site suppressor mutations measured using a thermal shift assay (see Materials and methods) with purified proteins. Refer to *Figure 4—figure supplement 2* for the thermal unfolding profiles of parent inactive mutants and (Parent inactive mutant, suppressor) pairs. Error bars represent standard deviation from at least two independent experiments. The results have been summarized in *Supplementary file 2*.

The following figure supplements are available for figure 4:

**Figure supplement 1.** Yeast surface display titrations of parent inactive mutants and (Parent inactive mutant, suppressor) pairs to determine $K_d$ (dissociation constant between CcdB displayed on the yeast surface and purified Gyrase).

**Figure supplement 2.** Thermal stabilities of purified CcdB Wild Type (WT), R10G, parent inactive mutants and (Parent inactive mutant, suppressor) pairs measured by thermal shift assay (TSA).

compromised ability to bind to its antitoxin CcdA results in increased toxicity in native contexts where CcdB and CcdA are both present. Thus, to maintain homeostasis in the system, evolutionary pressure defines a trade-off between function and stability of the protein (*Schreiber et al., 1994*), settling on an optimally stable wild type protein rather than a maximally stable one. This explains why the R to G mutation is not found in naturally occurring CcdB homologs. The screen employed in this work identifies stabilizing variants which are functionally competent to bind to only one of the binding partners, DNA Gyrase, explaining the identification of the R10G like mutation. To understand the molecular mechanism(s) by which R10G rescues L36A present at the core, we examined if any long range functional interaction was predicted between the sites by the program SCA (*Halabi et al., 2009*). No interaction was seen although this analysis was limited by low sequence diversity of CcdB. Experimentally, we observed that R10G stabilized the WT protein and other parent inactive mutants while the E11R substitution stabilized the parent inactive mutant V20F. The large conformational flexibility of glycine (in case of R10G) might stabilize the loop harboring residue 10 by accessing conformations not accessible to other residues. Further experiments need to be done to understand the mechanism of stabilization of the parent inactive mutant V20F by E11R and to determine if E11R, like R10G also functions as a global suppressor.

## ContactScore as a model discriminator

A decoy set of 10,659 models (*Adkar et al., 2012*) (doi:10.5061/dryad.3g092) of CcdB was used to probe the utility of the experimentally obtained contact information in model discrimination. The decoy set contained models ranging from 1.9–20.4 Å (backbone RMSD relative to the crystal structure, PDB id 3VUB [*Loris et al., 1999*]). The models were scored based on ContactScore, which was defined as the number of times the experimentally identified residue contact pairs (6 pairs) are within a cutoff distance of 7 Å of each other in a given model (*Figure 5A*) (see Materials and methods). ContactScore is an integral value ranging from 0 to 6, as there are six (Parent inactive mutant, proximal suppressor) pairs. Proximal suppressor mutants are likely to have their side chains facing towards the corresponding parent inactive mutant (*Figure 3B–F*), and hence, the side chain centroids of the pair are likely to be closer than their corresponding Cα atoms. Results for other choices of distances are shown in *Figure 5—figure supplement 1*. The distribution of recovery of models (defined as the percentage of models selected by the metric within a specified RMSD range, in a pool of models) with respect to their backbone RMSD relative to the crystal structure (*Figure 5—figure supplement 2*) shows the sensitivity of ContactScore and its relevance as a metric for model discrimination. Models satisfying all the experimental constraints that is ContactScore=6 are distributed in the RMSD bin <4 Å. The distribution progressively shifts towards a higher RMSD range with decrease in the number of constraints being satisfied (ContactScore<6). This emphasizes the sensitivity and selectivity of the metric. The correlation coefficient of a plot of RankScore as a function of residue depth in a model, $r_{depth}^{RankScore}$ (or) $r_{depth}^{score}$ has been previously used as a model discriminator (*Adkar et al., 2012*). $r_{depth}^{score} = 0.6$ for the native structure of CcdB. Thus, models with $r_{depth}^{score} \geq 0.6$ were selected as 'correctly folded models' when $r_{depth}^{score}$ was used as the metric. A comparison of the two metrics shows that ContactScore performs significantly better than $r_{depth}^{score}$, recovering 100%, 98% and 80% structures in backbone RMSD ranges 1.5–2 Å, 2–2.5 Å and 2.5–3 Å, respectively, while the latter recovered 0%, 7% and 12% in the above ranges (*Figure 5C*). ContactScore and $r_{depth}^{score}$ identified 585 and 67 models respectively with RMSD range <4 Å from the decoy set (*Figure 5A,B*). Thus, ContactScore is able to recover 66% of native-like models (RMSD <4 Å) from the dataset as compared to only 8% by $r_{depth}^{score}$.

We compared the decoy discrimination efficiency of $r_{depth}^{score}$ and ContactScore with another method which uses a simple scoring function based on residue accessiblity in globular proteins (*Bahadur and Chakrabarti, 2009*). The function (R$_s$) evaluates the deviation from the average packing properties of all residues in a polypeptide chain corresponding to a model of its three-dimensional structure (*Bahadur and Chakrabarti, 2009*). The parameter R$_s$ was calculated (see Materials and methods) for the CcdB decoy set. Since R$_s$ estimates deviation from the average Accessible Surface Area (ASA), the native structure should ideally possess the lowest value of R$_s$. However, when the CcdB decoy set was sorted according to the R$_s$ values, the native structure was ranked 934[th] and the correlation between RMSD and R$_s$ was seen to be only 0.3. These data demonstrate that both $r_{depth}^{score}$ and

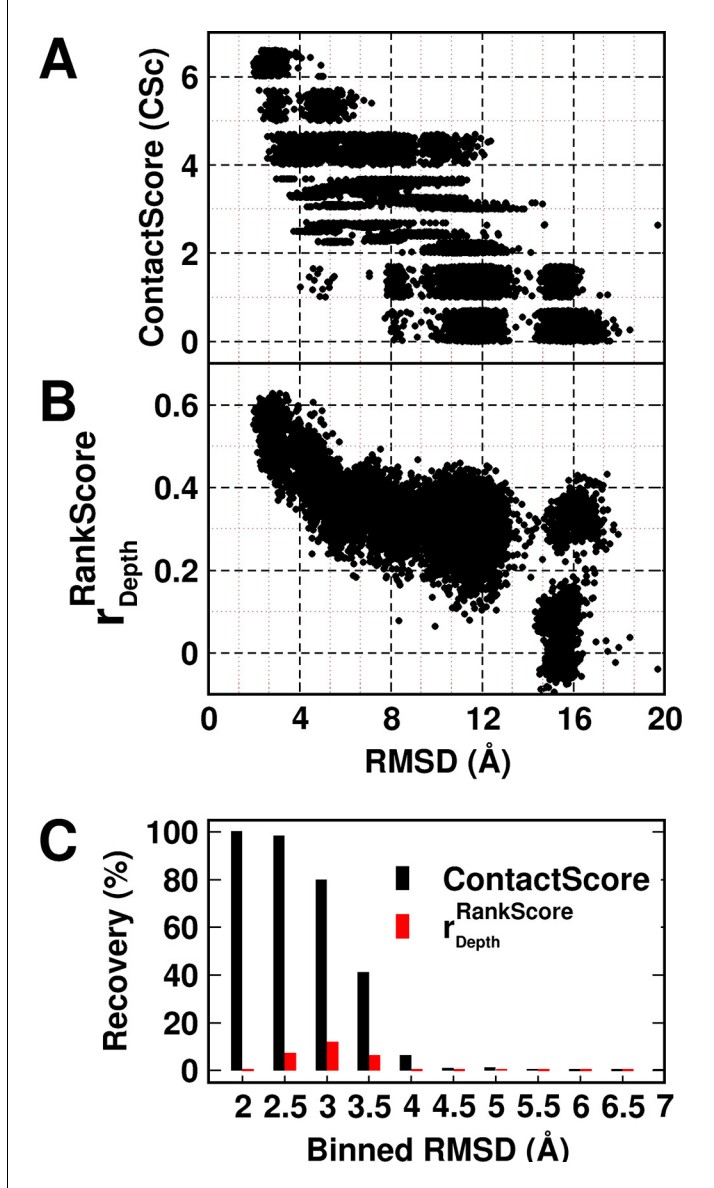

**Figure 5.** Experimentally determined 'ContactScore' and $r_{depth}^{RankScore}$ (**Adkar et al., 2012**) as model discriminators. Distribution of (**A**) ContactScore (CSc) and (**B**) $r_{depth}^{RankScore}$ as a function of backbone RMSD with respect to the crystal structure for 10,659 models of CcdB. CSc values are integers ranging from 0 to 6. Thus, models with different RMSD values (abscissa) can have the same CSc (ordinate). The Csc values for such points are randomly incremented by upto 0.7 units from their actual integral CSc value for clearer visualization. (**C**) Black and red bars represent recovery for CSc and $r_{depth}^{RankScore}$ respectively. CSc performs significantly better than $r_{depth}^{RankScore}$, recovering a much higher fraction of low RMSD models. The total RMSD range was divided into 0.5Å bins. Cutoff values of CSc and $r_{depth}^{RankScore}$ used were 6 and 0.6 respectively. Refer to **Figure 5—figure supplement 1** for recovery of CcdB models using Csc=6 shown for various choices of cut-off distances used to define contact. Refer to **Figure 5—figure supplement 2** for recovery of CcdB models as a function of different Csc values. These two plots demonstrate that a distance of <7 Å and Csc=6 are optimal for model recovery in this system.

The following figure supplements are available for figure 5:

**Figure supplement 1.** Recovery of CcdB models using a ContactScore (CSc) value of Csc=6 shown for various choices of cut-off distances used to define contact.

**Figure supplement 2.** Recovery of CcdB models as a function of different ContactScore (Csc) values using a fixed cut-off contact distance value of <7 Å.

ContactScore parameters derived from mutational data perform better than simple solvent accessibility based correlations such as the one observed above.

## Application of suppressor methodology to identify the functional conformation of the membrane protein DgkA in-vivo

The structures of the integral membrane protein, DgkA solved by X-ray crystallography (*Li et al., 2013*) and NMR (*Van Horn et al., 2009*) are different from each other in important respects. The NMR structure appears to be in a domain swapped conformation relative to the crystal structure. Several pairs of differential contacts serve to discriminate the two structures, including contacts made by residues V62, M66, I67, V68 and W112 (*Table 2*, *Supplementary file 3*, *Figure 6A, B*, *Figure 6—figure supplement 1*, see Materials and methods). Consequently, residues in proximity to each of these residues in the X-ray structure (PDB id 3ZE5 [*Li et al., 2013*]) are distant in the NMR structure (PDB id 2KDC [*Van Horn et al., 2009*]). Thus by constructing parent inactive mutants at the above positions and isolating corresponding suppressors, it should be possible to determine which of the two structures represents the functional conformation in-vivo. It has previously been shown (*Raetz and Newman, 1978*; *1979*) that cells deleted for *dgkA* do not grow under conditions of low osmolarity, providing a facile screen for both parent inactive mutants and their corresponding suppressors.

Each of the above five residues was individually randomized and the resulting small libraries were screened for inactive mutants. Charged and aromatic substitutions were excluded from the finally selected parent inactive mutants as these are likely to be highly destabilizing, making it difficult to isolate suppressors for such mutations. Indeed, in the case of CcdB it was more challenging to find suppressors for the aromatic parent inactive mutants (V18W, V20F), relative to other parent inactive mutants (such as L36A and L83S) (*Supplementary file 1*). V62Q, M66S, M66L, I67V, V68G and W112V were identified as parent inactive mutants from screening of single-site saturation mutagenesis libraries constructed at these positions. For these mutants, colonies appeared on plates only at high osmolarity (NaCl concentrations of 0.15%, 0.15%, 0.15%, 0.03%, 0.15% and 0.15% respectively) after 12 hr of incubation at 37°C, as opposed to cells expressing WT DgkA which grew even at 0% NaCl. Second-site suppressor mutagenesis libraries in which each residue in contact with the parent inactive mutants in both NMR and crystal structures was individually randomized, (*Table 2*) were screened for growth under low salt conditions.

At all selected parent inactive mutants, suppressors were found only at those positions in contact with the parent inactive mutant in the X-ray structure (i.e. V62Q/A41G, M66L/V38A, M66S/G35A, I67V/I103L, I67V/A104T and V68G/A100V), (*Figure 6*, *Figure 6—figure supplement 2*, *Table 2*,

**Table 2.** Experimentally determined (Parent inactive mutant, Suppressor) pairs for DgkA are spatially close only in the corresponding crystal structure.

| Parent inactive mutant position | Contact partners[a] in X-ray structure[b] | Contact partners[a] in NMR structure[c] | Experimentally identified (Parent inactive mutant, Suppressor) pairs |
|---|---|---|---|
| V62 | A41, 108, W112 | L102, I103 | (V62Q, A41G) |
| M66 | F31, G35, V38 | A99 | (M66L, V38A) (M66S, G35A) |
| I67 | A100, I103, A104 | F31, E34 | (I67V, I103L) (I67V, A104T) |
| V68 | A100, V101, A104 | F31, G35 | (V68G, A100V) |
| W112 | A41, I44, L58, S61 | - | [d] |

[a]Inter-helical contact pairs (see Materials and methods for details)

[b]X-ray structure of DgkA (PDB id 3ZE5 [*Li et al., 2013*])

[c]NMR structure of DgkA (PDB id 2KDC [*Van Horn et al., 2009*])

[d]No suppressors for this parent inactive mutant could be isolated, probably because the large volume change in the parent inactive mutant (W112V) is difficult to compensate by a single suppressor mutation.

[-]No inter-helical contacts

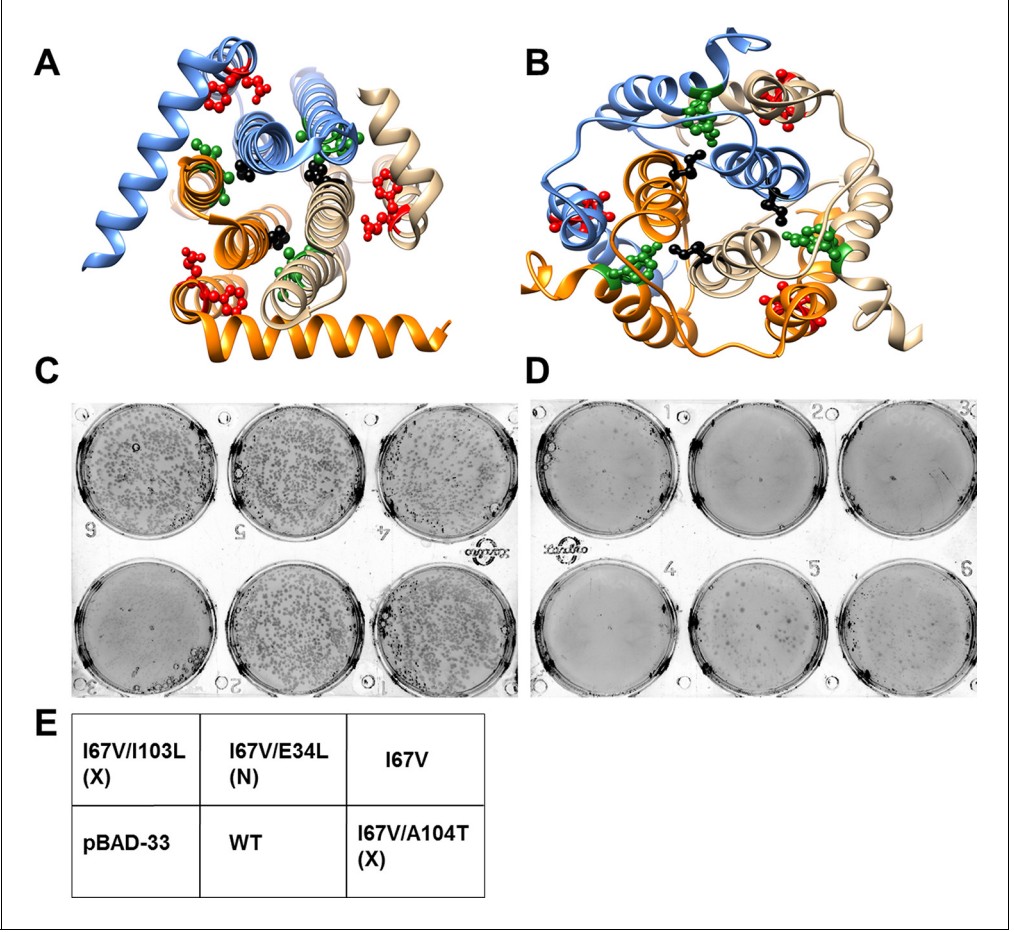

**Figure 6.** Screening of suppressors for the parent inactive mutant I67V DgkA. I67 (ball and stick in black) and inter-helical residues in contact with it in either (A) crystal (PDB id 3ZE5 [*Li et al., 2013*]) or (B) NMR (PDB id 2KDC [*Van Horn et al., 2009*]) structures of the homotrimeric DgkA are shown. Each monomer of the trimer is shown in a different color. The residues in proximity to I67 in a specific structure are shown in green, while the ones in contact in the alternate structure are highlighted in red. Differential contact pairs for other parent inactive mutants of DgkA have been shown in *Figure 6—figure supplement 1*. The figure has been prepared using the UCSF Chimera package (developed by Resource for Biocomputing, Visualization, and Informatics at the University of California, San Francisco [*Pettersen et al., 2004*]). Second-site suppressor libraries of the parent inactive mutant were constructed by randomizing each of the residue partners present in either of the two structures of the protein. Active mutants were screened on selective media at decreasing NaCl concentrations in an *E. coli* strain knocked out for *dgkA*. (C) and (D) show the phenotype of the putative suppressors isolated from the libraries at (C) 0.15% NaCl and 0.01% arabinose and (D) 0% NaCl and 0.01% arabinose. Phenotypes of suppressors corresponding to other parent inactive mutants have been shown in *Figure 6—figure supplement 2*. (E) A representative plate showing the location of the variants and the controls. '(X)' and '(N)' indicate the structure from which the residue partner has been chosen that is either X-ray or NMR structure, respectively. Parent inactive mutant I67V, WT DgkA and the empty vector pBAD-33 act as reference, positive and negative controls, respectively. The true suppressors are anticipated to grow on the plate with low salt concentration while the parent inactive mutant and (Parent inactive mutant, non-suppressor) pairs fail to grow. Suppressors (I67V, I103L) and (I67V, A104T), which are in spatial proximity in the crystal structure restore the growth defect of the parent inactive mutant I67V, whereas the (I67V, E34L) pair which is in proximity only in the NMR structure, fails to restore the growth defect.

The following figure supplements are available for figure 6:

**Figure supplement 1.** Differential contact residue pairs mapped onto the structures of DgkA.

**Figure supplement 2.** Screening for suppressors of parent inactive mutants of DgkA.

*Video 2*). The only exception was for the parent inactive mutant W112V, where no suppressors were experimentally identified, possibly due to the large change in volume for the parent inactive mutant relative to the WT residue. The data reported here are consistent results obtained from more than five independent experiments for each mutant. The results strongly suggest that the crystallized conformation is the native, functional conformation in-vivo.

## Computational approaches to predict spatially proximal residues

In the recent past there have been several computational efforts to identify residues in contact, involving analysis of correlated substitution patterns in a multiple sequence alignment of a protein. DCA (*Morcos et al., 2011*), PSICOV (*Jones et al., 2012*; *Nugent and Jones 2012*), GREMLIN (*Kamisetty et al., 2013*), SCA (*Halabi et al., 2009*) and EVfold (*Marks et al., 2011*) analyze co-variation matrix data from a multiple sequence alignment to deduce residues in contact. The methods rank residue pairs based on a co-variation or correlation score specific to each method. The top ranked pairs which typically lie in the top L/2 pairs, where L is the length of the protein sequence (*Jones et al., 2012*) are predicted to be in contact. The methods perform well when the size of the multiple sequence alignment is large, that is >5L (*Kamisetty et al., 2013*). DgkA (121 residues per protomer) exhibits large sequence diversity (4175 sequences in the multiple sequence alignment). Putative contact predictions by the above methods were analyzed by calculating sidechain-sidechain centroid distances between the predicted pairs using both X-ray and NMR structures of DgkA. Some high scoring, co-varying pairs predicted by DCA, GREMLIN, PSICOV and EVfold were found to be true contacts (centroid-centroid distance <7 Å, *Figure 7*) when mapped onto the crystal structure. There were a few high scoring pairs which were either far apart in the X-ray structure (predictions by PSICOV) or were in proximity when analyzed with the NMR structure (predictions by GREMLIN, PSICOV and EVfold). However, overall sequence co-variation data are more consistent with the X-ray structure, in agreement with conclusions from suppressor mutagenesis. Of the six contacts identified from our suppressor analyses (*Table 2*), three (62–41, 67–104, 68–100) were predicted in the top L/2 co-varying pairs by GREMLIN, PSICOV and EVfold (*Figure 7*), only 67–104 was predicted by DCA and none by SCA. This suggests that natural sequence co-variation and suppressor mutagenesis can provide complementary information.

The co-variation prediction methods use a multiple sequence alignment as input. The predictions therefore are not specific to the identities of the side-chains of the residues present in the sequence of interest at the predicted contact positions. Therefore, we also analyzed the predictions by calculating the Cα-Cα distances between the predicted pairs using both X-ray and NMR structures (*Figure 7—figure supplement 1*). No side chain information is involved in these calculations. Similar results were obtained as when using the sidechain-sidechain centroid distances. Co-variation prediction becomes increasingly challenging for proteins with very few homologs. CcdB (101 residues per protomer) has only 350 sequences (<5 L, where L is the length of the protein) in the multiple sequence alignment. Therefore, co-variation predictions for CcdB were not included.

## ΔΔG calculations for local suppressors of CcdB

There is considerable interest in accurate prediction of mutational effects on the free energy of folding (*Guerois et al., 2002*; *Kellogg et al., 2011*; *Shen and Sali, 2006*). We therefore examined whether ΔΔG calculations could be used to rationalize the identity of the experimentally observed local suppressors. To this end the difference in stability between the (Parent inactive mutant, suppressor) pair and the parent inactive mutant for CcdB mutants was calculated. $\triangle\triangle G_{folding}$ ($\triangle G_{folding}$Double mutant-$\triangle G_{folding}$-Parent inactive mutant) was calculated using

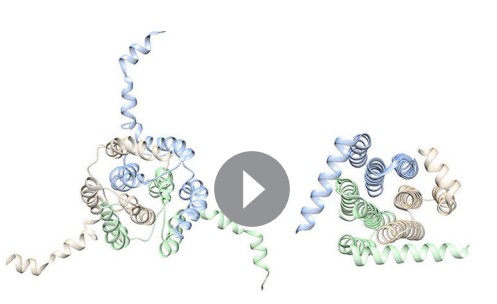

**NMR (left, PDB id 2KDC) and X-ray (right, PDB id 3ZE5) structures of DgkA**

**Video 2.** Experimentally obtained (Parent inactive mutant, suppressor) pairs mapped onto the two structures of DgkA (NMR PDB id 2KDC and X-ray PDB id 3ZE5) exhibit spatial proximity only in the corresponding crystal structure. Parent inactive mutant is abbreviated as PIM in the video.

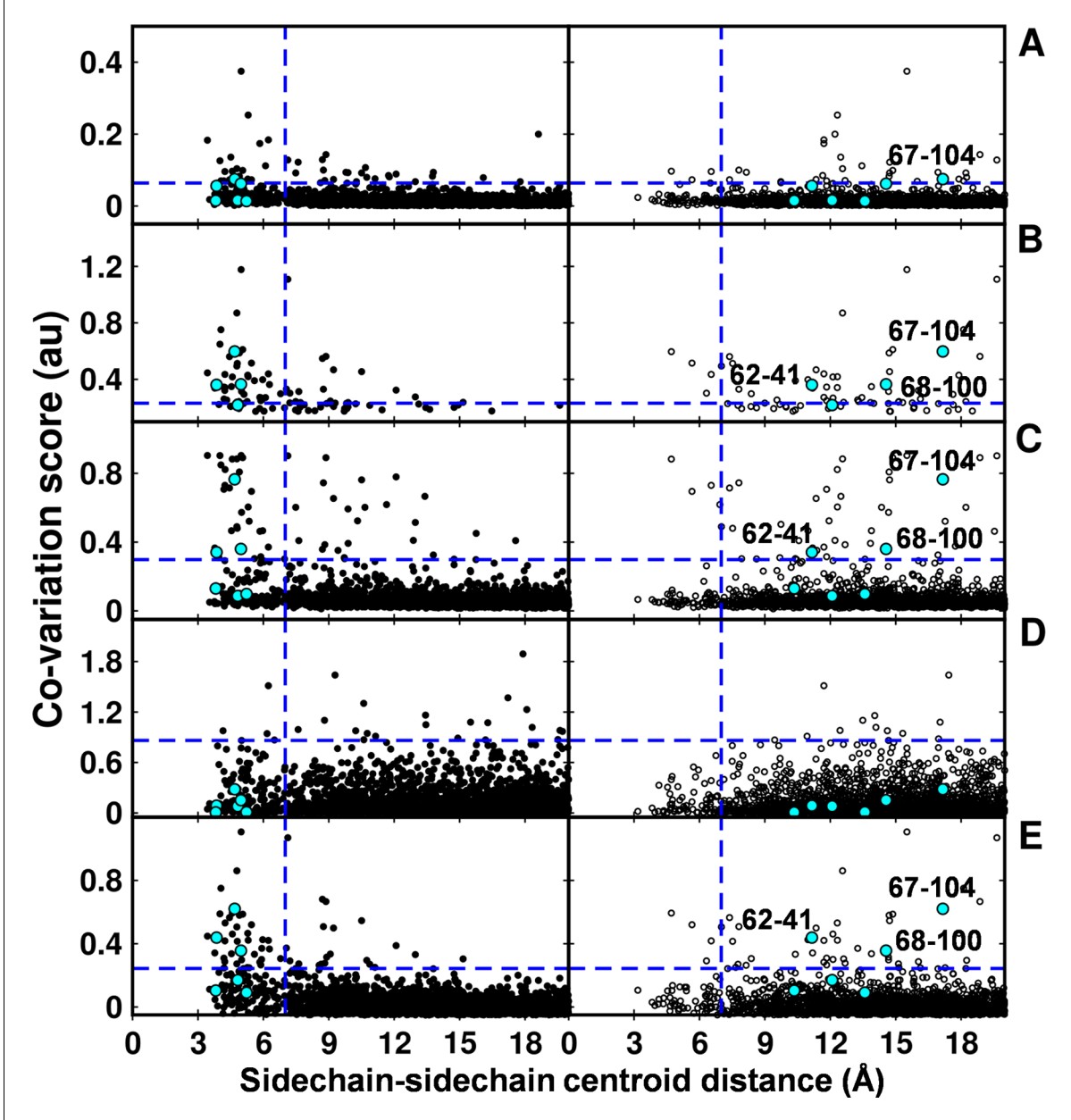

**Figure 7.** Computational analyses of co-varying residues for DgkA and comparison with experimentally determined contact residue pairs. Co-variation analyses using (**A**) DCA, (**B**) GREMLIN, (**C**) PSICOV, (**D**) SCA and (**E**) EVfold are shown. Residue pair side-chain centroid distances calculated from DgkA crystal (PDB id 3ZE5) and the NMR 1st pose structure (PDB id 2KDC) are shown in left (filled circles) and right (open circles) panels respectively. 'au' denotes arbitrary units. Blue lines parallel to the X-axis indicate the co-variation score of the L/2th residue pair (when arranged in descending order of the score), where L is the length of the protein (L=121 for DgkA). Blue lines parallel to the Y-axis indicate a sidechain–sidechain centroid distance of 7 Å between the predicted co-varying residue pairs in the corresponding structure. Experimentally determined spatially proximal (Parent inactive mutant, suppressor) contact pairs are shown in cyan. Although several computationally predicted contact pairs in the top L/2 predictions (top panel) are proximal to each other in the crystal structure, there are several predicted pairs which are far apart. Computational analyses of co-varying residues for DgkA and comparison with experimentally determined contact residue pairs using Cα-Cα distances instead of sidechain-sidechain centroid distances are shown in *Figure 7—figure supplement 1*.

The following figure supplement is available for figure 7:

**Figure supplement 1.** Computational analyses of co-varying residues for DgkA and comparison with experimentally determined contact residue pairs using Cα-Cα distances instead of sidechain-sidechain centroid distances.

Rosetta v3.3 (*Kellogg et al., 2011*). Putative proximal suppressors were considered to arise at residues within 7 Å (sidechain–sidechain centroid distance) of the parent inactive mutant. Many stable substituents were predicted ($\triangle\triangle G_{folding}$<0, *Figure 8*). However, amongst the six experimentally identified stable compensatory pairs, only L36A/M63L (−3.7 kcal/mol) was predicted to be stable. The remaining five contact pairs were predicted to be either marginally stable or unstable. Several other mutations besides the experimentally determined ones were predicted to be stabilizing for example V5F/L16G, V18W/I90A, V20F/I90A, L36A/V54I and L83S/V18I. These might be present in the earlier rounds of sorting but are lost in later rounds due to stringent sort conditions. A marked bias for aromatic substitutions was observed in the predictions (*Figure 8*, substitutions underlined in magenta) though such aromatic substitutions were not observed experimentally. Aromatic substitutions are rigid and were found to over pack the cavity created by the parent inactive mutants in the models generated using Rosetta. Further, several of the mutations that were computationally predicted to be highly stabilizing are unlikely to be so as they are not complementary in size to the original parent inactive mutant, for example L36A/W61F, V5F/L16Y, V18W/I90F and V20F/I90F. If aromatic substitutions are excluded, Rosetta predictions using ΔΔG values are in reasonable qualitative agreement with experiment.

A similar analysis was done using FoldX (*Guerois et al., 2002*) (*Figure 8—figure supplement 1*). However, these predictions were in poorer agreement with the experimental results, compared to those of Rosetta. Thus, in addition to their use in protein structure prediction, results from such suppressor analyses can also be used to benchmark and improve computational approaches to predict mutational effects on protein stability.

## Discussion

Interactions at the protein core are important in determining its structure and stability. The saturation-suppressor mutagenesis methodology described here (*Figure 1*, *Figure 1—figure supplement 3*) enabled identification of 8 residues at the hydrophobic core and their pairwise interactions (*Figure 3A,F*) placing important constraints on packing of the model protein CcdB. Encouragingly, proximal suppressors could be obtained for four of the five parent inactive mutants in the case of CcdB as well as DgkA. There may be mutations in the core which allow the protein to fold but are functionally defective (*Figure 1—figure supplement 2*) (*Roscoe et al., 2013*). We eliminated such mutations as the CcdB screen required variants to both fold into a stable conformation and be functionally active to bind to Gyrase. The experimentally identified residue pairs for CcdB are in physical contact and have suppressor positions common to each other (*Table 1*, *Figure 3B–F*, *Video 1*). This interlinked network of residues restricts the conformational space during folding. The ContactScore metric defined above selects models satisfying the identified spatial constraints. A histogram of recovery of models with respect to backbone RMSD of the models selected by this metric shows a maximum of 100% for models with backbone RMSD <2 Å, gradually decreasing to 66% for models <4 Å and subsequently plateauing to 0% for models >5 Å, reflecting the sensitivity and accuracy of the parameter. A high recovery is important for protein model discrimination, as typically there will be very few low RMSD models in the candidate set of predictions. The ContactScore metric also performs better than a simpler approach based on deviation of residue accessibilities in a model from their average values in a large dataset of proteins (*Bahadur and Chakrabarti, 2009*) though an exhaustive comparison with other metrics for model discrimination has not been carried out. The efficiency of model discrimination using the ContactScore metric will likely be a function of quality of the decoy dataset as well as the accuracy of the contacts inferred from suppressor mutagenesis. In a dataset where very few low RMSD models are present or where not all contacts identified are true contacts, the model recovery will be lower than for the CcdB decoy dataset described here. Further, the small number of (Parent inactive mutant, suppressor) pairs identified here will not be sufficient for larger proteins, thereby affecting the model discrimination efficiency. However, most of these limitations can be overcome by identifying a higher number of (Parent inactive mutant, suppressor) pairs for the protein of interest and by combining residue contacts inferred from saturation mutagenesis with other experimental or computational constraints.

The saturation suppressor mutagenesis approach was extended to the important case of membrane proteins. Many membrane proteins adopt multiple conformations (*Tokuriki and Tawfik, 2009*) and membrane mimetics used to solubilize and stabilize membrane proteins can affect their

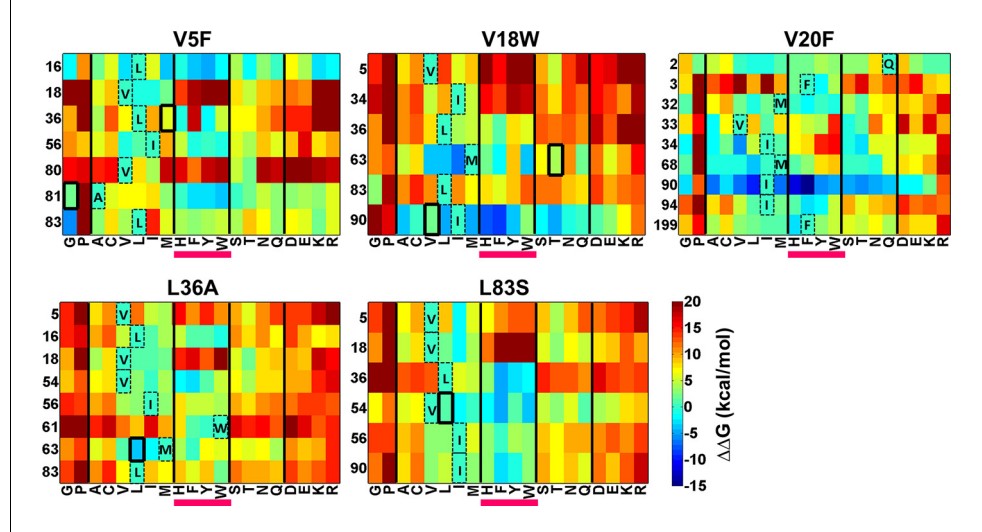

**Figure 8.** Heatmaps showing calculated values of $\Delta\Delta G_{folding}$ using Rosetta for double mutants of CcdB. All 19 mutations (X-axes) were made at positions (Y-axes) whose side chain centroids are within 7 Å of the side chain centroid of the corresponding parent inactive mutant. The parent inactive mutants are indicated in bold on the top of each heatmap. For example, the bottom left corner of the first panel represents the $\Delta\Delta G$ value for the (V5F, L83G) (Parent inactive mutant, suppressor) pair. Position 199 (98+101) in the V20F heatmap refers to residue 98 of the other protomer. Wild type residues are shown in dashed boxes in single letter code. Experimentally obtained suppressors are indicated by thick black boxes. Double mutants showing negative values of $\Delta\Delta G_{folding}$ (blue shades) are ones where the putative suppressor mutation is predicted to have a stabilizing effect on the inactive mutant. Aromatic substitutions are underlined in magenta. Residues on X-axis are grouped into the following classes, separated by thick vertical lines: (G,P), aliphatic (A,C,V,L,I,M), aromatic (H,F,Y,W), polar (S,T,N,Q) and charged (D,E,K,R).Similar analysis was performed using FoldX (*Figure 8—figure supplement 1*).

The following figure supplement is available for figure 8:

**Figure supplement 1.** Heatmaps showing calculated values of $\Delta\Delta G_{folding}$ using FoldX for double mutants of CcdB.

conformations (*Cross et al., 2013*). There are two reported structures of the integral membrane protein DgkA, one solved by X-ray crystallography (*Li et al., 2013*) and one by solution NMR (*Van Horn et al., 2009*) which differ significantly from each other. Using our suppressor methodology, we unambiguously identified six residue-residue contacts which were all present in the crystal structure but were spatially distant in the NMR structure. This suggests that the conformation of the protein in the lipidic (monoacylglycerol) cubic phase conditions (used in X-ray crystallography [*Li et al., 2013*]) is the functional conformer in-vivo and is not an artifact resulting from the presence of thermostabilizing mutations and minor distortions that might result from crystal contacts. Our results are also consistent with a recent reanalysis of oriented sample solid state NMR data for the protein in liquid crystalline bilayers (*Murray et al., 2014*) which showed better overall agreement with the crystal than with the solution NMR structure.

Broad application of suppressor methodology to systems where no structural information is available requires accurate discrimination of buried from exposed active-site residues and of distal from proximal suppressors. As discussed above, buried and active-site residues can be distinguished based on their mutational sensitivity patterns as well as from data on mutant protein levels, sensitivity to chaperone overexpression patterns (*Tokuriki and Tawfik, 2009*) and residue conservation patterns (*Melamed et al., 2015*). Distal suppressors are likely to be on the surface (*Bank et al., 2015*). If the correlation of mutational sensitivity/RankScore with depth seen for CcdB holds for other globular proteins, this should allow distinction of local and global suppressors in such systems, as long as the majority of global suppressors lie on the surface. However, unambiguous distinction between global and local suppressors will likely not always be possible with a limited number of (Parent inactive mutation, suppressor) pairs. Also, in some cases, a single suppressor mutation may not be

sufficient to restore the activity of a parent inactive mutant. In such cases, the network of correlating residues will be more complex and additional experimental data will be required to identify a set of mutations which will suppress a parent inactive mutant.

Unlike globular proteins, little is known about sensitivity to mutation in membrane proteins or natively unfolded proteins. Until such data becomes available it will be challenging to apply this methodology to these systems. In the case of DgkA, we had the much simpler objective of distinguishing between two possible structures. While distinguishing between global and local suppressors maybe more challenging in membrane proteins, given sufficient double mutant data, it should be straightforward because each global suppressor should suppress a much larger number of parent inactive mutants than all local suppressors. The average relative frequency of obtaining proximal versus distal suppressors is currently unknown. If a single-site saturation mutagenesis library is enriched for inactive mutants (Parent inactive mutants), subjected to random mutagenesis and screened for suppressors, the resulting population will be enriched for global suppressors (*Bershtein et al., 2006*; *2008*). This is because a global suppressor will suppress multiple parent inactive mutants. However, for a specific parent inactive mutant, it is not obvious that global suppressors will dominate. A recent study examined a library of the 75 amino acid RRM domain of the yeast poly-A binding protein (*Melamed et al., 2013*). Functional scores for 1246 single and 39,912 double mutants were obtained. Epistatic interactions were enriched for residue pairs with short sequence spacing (<5) and short distance (10–15Å). Another recent study reported exhaustive screening of single and double mutants of GB1 (*Olson et al., 2014*). The majority of pairs displaying positive epistasis had Cβ-Cβ distances <8 Å. Both of the above studies indicate that local suppressors may occur at a higher frequency than global ones with respect to individual parent inactive mutants but more data is required to confirm this.

Experimental approaches discussed previously to identify second-site suppressors used random mutagenesis and/or directed evolution to generate suppressor libraries. Although these libraries have high diversity and mutations at multiple residue positions, they typically do not have more than a single base substitution at any codon. It should be noted that single base changes can sample only 39% of all possible amino acid substitutions. Hence, they do not exhaustively sample all possible second-site suppressors for a given inactivating mutation. This is important, since for a given parent inactive mutant there appear to be only a few local suppressors, and these could well be absent in a library generated by conventional random mutagenesis.

Computational approaches to identify spatially proximal residues require a large number of homologous sequences to be present (typically greater than five times the length of the protein) (*Kamisetty et al., 2013*). These approaches do not work well for proteins like CcdB due to the limited evolutionary diversity in the multiple sequence alignment. Another constraint is that the range of substitutions sampled at a given position during natural evolution is limited by functional and stability constraints, that can be relaxed in a laboratory setting. For the protein DgkA, for which there are several sequences, the results of the five computational approaches to identify co-varying residues were more consistent with the crystal structure of DgkA than with the NMR structure. Several of the contacts identified by our suppressor approach were not identified by the sequence co-variation. Since our experimental strategy identifies only a few (Parent inactive mutant, suppressor), a direct comparison of these results with predictions by computational methods is not possible. However, our approach, provides complementary information to these existing methods, and can be usefully combined with them to guide protein structure prediction. The approach described here uses saturation-suppressor mutagenesis to identify spatially proximal residues. The library generation design adopted here for CcdB, constructs the second-site saturation library in the background of individual inactive mutants chosen from a single-site saturation mutagenesis library comprising of ~1430 mutants of CcdB (*Adkar et al., 2012*).

Second-site suppressors can be either proximal or distal to the mutation site. Virtually all suppressors identified in the study (for CcdB) increased the thermal stability relative to the original parent inactive mutant. Proximal suppressors are likely to ameliorate packing defects and hence restore stability. These have previously been reported to restore activity (*Machingo et al., 2001*), increase thermal stability and restore packing (*Pakula and Sauer, 1989*). Residues distant from the site have previously been found to function by either increasing global thermodynamic stability (*Araya et al., 2012*; *Bershtein et al., 2008*; *Pakula and Sauer, 1989*), increasing activity relative to the wild type protein without any substantial increase in stability (*Hecht and Sauer, 1985*) or improving foldability

without much effect on the thermodynamic stability (*Sideraki et al., 2001*). In the present study, local suppressors could be obtained for each of the five parent inactive mutants in CcdB and five of six parent inactive mutants in DgkA, regardless of the location and nature of the parent inactive mutant. The distal suppressor R10G in CcdB increased the $T_m$ by 8°C, relative to WT CcdB (*Supplementary file 2*) which rescues the destabilized mutant L36A.

Global suppressors have been shown to often comprise of consensus/ancestral mutations (*Bershtein et al., 2008*). We analyzed the consensus/ancestral mutations for CcdB. The consensus sequence was obtained from a multiple sequence alignment of 350 homologs using MATLAB and the ancestral sequence was obtained using the FastML server (*Ashkenazy et al., 2012*). The likely global suppressors we obtained in the present study are R10G and E11(R/K/P). At R10 the ancestral and consensus amino acids are P and R respectively and at E11 they are A and N respectively. Hence, at least for these two positions, the ancestral/consensus amino acids were different from the experimentally obtained suppressors. Further experiments are required to ascertain whether the ancestral/consensus amino acids will also act as global suppressors.

Advances in the field of protein structure prediction integrate various computational approaches with distance restraints derived from cross-linking experiments and mass spectrometry (*Young et al., 2000*), sparse NOE data (*Bowers et al., 2000*; *Li et al., 2003*; *Thompson, et al., 2012*), residual dipolar coupling data (*Haliloglu et al., 2003*; *Qu et al., 2004*), chemical shift data (*Shen et al., 2008*) from NMR experiments, co-varying residues identified from statistical analysis of genomic data (*Hopf et al., 2012*; *Jones et al., 2012*; *Ovchinnikov et al., 2014*; *Ovchinnikov et al., 2015*; *Sulkowska, et al., 2012*) to determine structure. The combination of site-directed mutagenesis or doped oligonucleotide based synthetic library generation strategies with deep sequencing has expanded our understanding of sequence, structure, function relationships (*Tripathi and Varadarajan, 2014*). High resolution mutational analyses using single-site saturation mutagenesis have facilitated understanding the influence of each residue on a protein's structure, stability, activity, specificity and fitness (*Adkar et al., 2012*; *Fowler et al., 2010*; *Roscoe et al., 2013*; *Tripathi and Varadarajan, 2014*). Expanding this landscape by integrating spatial constraints isolated from paired mutational phenotypes can greatly advance our efforts to construct highly accurate structural models especially where evolutionary information is sparse, because of the paucity of homologous sequences. Unbiased examination of all pairwise mutant combinations is currently not feasible because the large library size cannot easily be sequenced at sufficient depth using deep sequencing ($^{100}C_2 \times 400$ or $2 \times 10^6$ for a 100 residue protein [*Tripathi and Varadarajan, 2014*]). However, with continuous improvements in deep sequencing and mutant generation technologies this is likely to change soon. The methodology outlined here can be similarly applied to any protein or protein complex where mutation can be coupled to a phenotypic readout. In the two systems studied here, we identified intramolecular suppressors of core packing defects, and active-site mutants were excluded from the set of parent inactive mutants. However, a similar approach can be used to obtain structural information on biomolecular protein:protein complexes. In this case, single-site saturation mutagenesis libraries at active-site residues of one partner would be screened against parent inactive mutants located, at active-site residue of the other partner to identify inter-molecular suppressors. These in turn would provide constraints to build structural models of the complex.

In these proof of principle studies, mutant identities were determined after single (DgkA) or multiple (CcdB) rounds of screening, using Sanger sequencing. In the case of CcdB, residue burial was accurately inferred exclusively from mutational data. Proximal and distal suppressors could be accurately distinguished based on their mutational sensitivity when present as single mutants and all proximal suppressors were in close contact with the corresponding parent inactive mutant. For other systems it remains to be seen how well this approach will work and how many suppressors will be required for accurate model discrimination. However, as with earlier studies using single mutant libraries (*Adkar et al., 2012*; *Fowler et al., 2010*; *Tripathi and Varadarajan, 2014*), enrichment of mutant pairs at each stage during selection or screening can be monitored using deep sequencing. Using this approach it will be possible to compare phenotypes of large numbers of partial loss-of function single mutants and corresponding double mutants. This data will help distinguish local from global suppressors and provide multiple constraints to guide macromolecular structure prediction and determination. These efforts may prove beneficial in resolving the gap between protein sequence and structure and also in the isolation of mutants with improved stability and foldability, relative to WT.

## Materials and methods

### A. CcdB

### Screening of CcdB single-site, saturation mutagenesis library in *E. coli*

This was described in (*Adkar et al., 2012*). Briefly, the CcdB gene was cloned under control of the $P_{BAD}$ promoter. Expression from this promoter is repressed by glucose and induced by arabinose in a dose-dependent manner ( *Guzman et al., 1995*). Since active CcdB is toxic to *E. coli*, cells expressing WT and WT-like mutants are killed even under highly repressed conditions. Partial loss of function mutants do not kill cells when expressed at low levels but show an active phenotype (cell death) when expressed at higher levels. The single-site saturation mutagenesis library was transformed into *E. coli* strain Top10pJAT and relative populations of surviving mutants at each of seven different expression levels were measured as follows. The CcdB gene from surviving colonies at each expression level was amplified and tagged with a Multiplex IDentifier sequence (MID) unique to each growth condition. The libraries were then subjected to deep sequencing.

### Definition of mutational sensitivity score (MS_seq), RankScore and $r^{score}_{depth}$

A mutational sensitivity score (MS_seq) was defined as the MID at which cells containing the mutant protein die. It can be deduced from the number of sequencing reads as the MID at which the number of reads for a particular mutant decreases by fivefold or more compared to the previous MID. The MS_seq values range from 2 to 9, two representing the least sensitive (most active) and nine the most sensitive (least active) mutants. The cumulative number of mutants active at each MS_seq value was calculated along with their percentage frequency in the population. Mutants with an MS_seq of two were assigned a rank of one (say, they constitute a% of the total mutant population). For mutants with MS_seq of three, a rank of a+1 was assigned (say b% of the total mutant population). For mutants with MS_seq of four, rank of b+1 was assigned. Similarly, ranks are assigned to the remaining mutants. RankScore is the average of assigned ranks of all the available mutants at a position. It combines the protein activity data at various expression levels. A lower RankScore indicates lower mutational sensitivity. It was shown (*Adkar et al., 2012*) that RankScore is correlated with residue depth in the protein structure (*Loris et al., 1999*). The correlation coefficient of RankScore as a function of depth is defined as $r^{score}_{depth}$.

### Construction of double mutant library

A single-site saturation mutagenesis library of CcdB (*Jain and Varadarajan, 2014*) was cloned in a Yeast Surface Display vector pPNLS (*Bowley et al., 2007*) between *SfiI* sites. In this background, various single-site mutants were made by three fragment recombination of gapped vector and two overlapping CcdB fragments. PCR to amplify the overlapping fragments of CcdB was carried out with Phusion DNA polymerase (Finnzymes) for 15 cycles with 1–5 pg template DNA. pPNLS vector containing a ~1 kb stuffer sequence was digested with *SfiI* (New England Biolabs, NEB) to remove the stuffer insert, and gel purified. The digested vector and the two CcdB fragments were transformed by three fragment homologous recombination in *S. cerevisiae* EBY100 (*Gietz and Schiestl, 2007*). Yeast cells were incubated for 3 hr at 30°C in 30 ml YPD broth after transformation. The cells were washed twice with 30 ml sterile water and grown in 500 ml SDCAA for 32 hr at 30°C, 250 rpm. The total number of transformants was estimated by plating a small amount of the transformed cell suspension (30 µl of 30 ml) on an SDCAA agar plate and multiplying the number by the dilution factor. Cells representing 100 times the total transformants obtained were re-grown in 300 ml SDCAA for 16-20 hr at 30°C, 250 rpm and stored in aliquots of $10^9$ cells in YPD (HiMedia) containing 25% glycerol at −70°C.

### Yeast surface display and selection of second site suppressor mutants

Inactive mutants and corresponding double mutant libraries cloned in pPNLS were displayed on the surface of yeast (*Chao et al., 2006*) at either 30°C (L36A, L83S parent and mutant libraries) or 20°C (V5F, V18W, V20F parent and mutant libraries) for 16 hr for all experiments related to yeast surface display described here. Binding of chicken anti-HA antibody (SIGMA, 1:300) to the HA tag at the N terminus of CcdB was used to monitor surface expression of the displayed protein. Goat anti-chicken IgG conjugated AlexaFluor-488 (Invitrogen, 1:300) was used as the secondary antibody. The cmyc

tag fused to the C terminus of CcdB in the pPNLS vector was removed to enable binding of the ligand Gyrase, which has a 3xFLAG tag at its C terminus. Gyrase binding was assessed by binding of mouse anti-FLAG antibody (SIGMA, 1:300) and rabbit anti-mouse IgG conjugated AlexaFluor-633 (Invitrogen, 1:1800) secondary antibody. Approximately $10^7$ double-labeled cells from each library were sorted on a BD FACS Aria-III flow cytometer (488 nm, 633 nm lasers for excitation and 530/30 nm, 660/20 nm bandpass filters, respectively, for emission) to enrich mutants which show better surface expression and binding than the reference parent inactive mutant. The concentration of Gyrase-3xFLAG used depended on the corresponding affinity of the reference inactive mutant. Equilibrium dissociation constant values ($K_d$) for the reference mutants were measured using a yeast surface display titration as described (*Chao et al., 2006*). Sorting of the libraries was carried out for multiple rounds till (i) analysis of the population grown after the final sort showed at least 10% of the population in a gate which contained 0% of the corresponding parent inactive mutant (*Figure 2*, panel showing L83S-lib after sort3), and (ii) there was no further improvement in signals from the enriched library with subsequent rounds of sort. The stringency of the sort was progressively increased by decreasing the concentration of Gyrase and gating the top ~1% of the population, which largely excluded the reference mutant (see *Supplementary file 1*).

## Identification of suppressor mutants

Yeast cells harvested from 25 ml saturated culture of sorted libraries were resuspended in buffer P1 (supplied with Qiagen plasmid miniprep kit) and vortexed in the presence of acid-washed glass beads (SIGMA) for 10 min. The suspension was incubated with Zymolyase (30U, G-Biosciences) at 37°C for 4 hr to break the cell wall. A Qiagen plasmid miniprep kit was used for further downstream processing of the cells to purify the plasmid. CcdB gene inserts amplified from the libraries were cloned into the pTZ57R/T TA vector using an InsTAclone PCR cloning kit (ThermoScientific) and transformed into *E. coli* XL1-Blue cells for blue-white screening (*Langley et al., 1975*). The CcdB inserts from 96 randomly picked white colonies derived from each library after the final round of sorting, were sequenced by Sanger sequencing at Macrogen, Korea, to identify second-site suppressors.

## Purification and thermal denaturation of CcdB mutants

The *ccdb* gene initially cloned in pPNLS vector was cloned into pBAD-24 bacterial expression vector (*Guzman et al., 1995*) by Gibson assembly (*Gibson et al., 2009*). *E. coli* CSH501 cells are resistant to the toxin CcdB due to mutation in the chromosomal copy of *gyrA* which eliminates binding between CcdB and DNA Gyrase; the strain was kindly provided by Dr. M Couturier (Universite Libre de Bruxelles, Belgium). These cells, transformed with pBAD-24-CcdB plasmid were inoculated in 200ml LB medium, grown till $OD_{600}$ 0.7 at 37°C, induced with 0.2% arabinose and grown for an additional 12 hr at 18°C. The cells were harvested at 4,000 rpm for 10 min at 4°C, resuspended in 25 ml resuspension buffer (0.05 M HEPES, pH 8.0, 1 mM EDTA, 10% glycerol), containing 200 µM PMSF. The cells were lysed by sonication on ice. The solution was centrifuged at 14,000 rpm, for 30 min, at 4°C. The supernatant was loaded onto the pre-equilibrated CcdA (residues 46–72) affinity column (prepared using Affigel-15, Bio-Rad, Hercules, CA, as per instructions in the manual), and incubated for 4 hr at 4°C. The column was washed thrice with 20 ml coupling buffer to remove the unbound protein and eluted with 0.2 M Glycine, pH 2.5 in an equal volume of 400 mM HEPES, pH 8.5 and concentrated using a Centricon centrifugal filter unit (Millipore, MW cut-off 3 kDa). Fluorescence based thermal shift assay was carried out on an iCycle iQ5 Real Time Detection System (Bio-Rad). 25 µl of the reaction mixture containing 4 µM CcdB purified protein, 25x Sypro orange dye and buffer (200 mM HEPES and 100 mM glycine, pH 7.5) was subjected to thermal denaturation on a 96-well iCycleriQ PCR plate, from 20°C to 90°C with an increment of 0.5°C/min. Denaturation of the protein was also carried out in the presence of 20 µM CcdA peptide (residues 46–72). Sypro orange binds to the exposed hydrophobic patches of a protein, leading to an increase in observable fluorescence of the dye as the protein unfolds (*Niesen et al., 2007*). The data was fitted to a standard four parameter sigmoidal equation $y = LL + ((UL-LL)/(1+e^{(Tm-T)/a}))$ using SigmaPlotv11.0, where y is the observed fluorescence signal, LL and UL are the minimum and maximum intensities respectively during the transition, a is the slope of the transition, $T_m$ is the melting temperature (midpoint of the thermal unfolding curve or the temperature at which 50% of the protein is unfolded) and T is the experimental temperature.

## Contact score

10,659 models for CcdB were generated as described (*Adkar et al., 2012*). Each model (m) was allotted a ContactScore defined as,

$$\mathrm{ContactScore}_m = \sum_{i=1}^{n} S_i(x,\,y)$$

where, S(x,y)=1 if the distance between the side chain centroids of residues x and y is <7 Å in the crystal structure of CcdB (PDB id 3VUB [*Loris et al., 1999*]) else S=0, n denotes the number of experimentally determined probable contact pairs, (x,y).

## Calculation of the $R_s$ parameter

The function (Rs) (*Bahadur and Chakrabarti, 2009*) evaluates the deviation from the average packing properties of all residues in a given structural model. $R_s$ is calculated using the following formula, where $ASA_{xi}$ and $<ASA_x>$ are the accessible surface area values of residue X at position *i* and the average ASA of residue X in a large dataset, respectively (*Bahadur and Chakrabarti, 2009*).

$$R_s = \sum_{i=1}^{whole\ chain} \frac{|ASA_{xi} - <ASA_x>|}{<ASA_x>}$$

The $R_s$ parameter was calculated for all the 10,659 models in the CcdB decoy set. The ASA was calculated using NACCESS (v2.1.1) (*Hubbard, 1992*).

## In-silico identification of suppressor mutants using ΔΔG calculations

All 19 possible mutations were computationally introduced at residue positions whose side chain centroids were within 7 Å of the side chain centroid of the chosen parent inactive mutant residues that is, V5F, V18W, V20F, L36A and L83S. The high resolution protocol for ΔΔG calculation in Rosetta version 3.3 (*Kellogg et al., 2011*) was used to calculate ΔΔG*folding*, where,

△△G *folding*=△G *folding* (Double mutant) - △G *folding* (Parent inactive mutant)

Double mutants showing negative values of △△G*folding* were predicted to stabilize the parent inactive mutant. The following command was used:

~/rosetta_source/bin/ddg_monomer.linuxgccrelease -in:file:s ~/min_cst_0.5.dimer_0001.pdb

-in::file:fullatom -ignore_unrecognized_res -constraints::cst_file~/input.cst -database ~/rosetta_database/ -ddg::mut_fileresfile -ddg::iterations 50 -ddg::weight_filesoft_rep_design -ddg::local_opt_only false -ddg::min_cst true -ddg::ramp_repulsive true -ddg::sc_min_only false -ddg::mean false -ddg::min true -ddg::dump_pdbs true.

BuildModel protocol of FoldX version 3.6 beta (*Guerois et al., 2002*) was also used to calculate ΔΔG*folding* for mutants. The side chains of the neighboring residues were optimized in order to accommodate the mutant side chain, without allowing backbone flexibility.

## B. DgkA

### Identification of differential contacts

In the present study, true contacts were identified using a sidechain-sidechain centroid distance ≤7 Å as the cutoff, for both X-ray and NMR structures of DgkA. The NMR structure (PDB id 2KDC) has coordinates for 16 poses. For the figures and distances listed in the paper we used only the first pose. However, we calculated the Root Mean Square Fluctuations (RMSF) for each residue (Cα atom) taking the average structure as the reference structure. Greater fluctuations are observed near the N-terminus of the protein (residues 1 to 33). Hence, we did not consider differential contacts within this region. Since, the region beyond the first 33 residues shows low RMS fluctuations, we report sidechain-sidechain centroid distances for residues showing differential contacts with the parent inactive mutant in the X-ray and the NMR first pose structure. In *Supplementary file 3*, we report the closest distance amongst all the 16 poses between each parent inactive mutant and all residues shortlisted in *Table 2*.

Residue pairs in contact in X-ray (PDB id 3ZE5 [*Li et al., 2013*]) and NMR (PDB id 2KDC [*Van Horn et al., 2009*]) structures of DgkA were identified using the CMA server (*Sobolev et al., 2005*) (http://ligin.weizmann.ac.il/cma/) with the default parameters and threshold value of 10 Å². Helix definitions mentioned in corresponding PDB header files were used to

eliminate the contact pairs present in the same helix. Parent inactive mutant positions (say, 'X') were defined as residues common to the identified contact residue pair subsets in both structures, but with different partners (say, 'Y') in the two structures. The following criteria were additionally used to select X, that is (i) charged residues (X) were removed (ii) X and Y should not be involved in side-chain hydrogen bonding with each other (iii) X should have $\geq$2 contact partners in at-least one of the structures (iv) Sequence separation between X and Y, X-Y >30 (v) $Y_{NMR} - Y_{X-ray} > 6$.

## Cloning, mutagenesis and isolation of parent inactive mutants and their suppresssors

The gene sequence corresponding to a cysteine-less (C46A/C113A) form of DgkA (referred as WT here) (*Van Horn et al., 2009*) along with an upstream 30 bp RBS sequence (CTCGAGCCCGGGGTCG-ACGGCTCTGCGGGC) was synthesized and cloned between *KpnI* and *PstI* sites in the pBAD-33 vector (*Guzman et al., 1995*), under the AraC promoter at GenScript. Selected parent inactive mutant positions were randomized by single site saturation mutagenesis using NNN codons, by inverse PCR (*Jain and Varadarajan, 2014*). The amplicon was purified, phosphorylated (T4 Polynucleotide Kinase, NEB), ligated (T4 DNA Ligase, NEB), transformed into *E. coli TOP10* cells, plated on SB (HiMedia) containing 2% NaCl, 34 μg/ml chloramphenicol and 50 μg/ml kanamycin and incubated at 37°C for 12 hr. *E. coli BW25113*, knocked out for the chromosomal copy of the *dgkA* gene (Keio collection (*Baba et al., 2006*), id JW4002-1;F-, Δ(*araD-araB*)567, Δ*lacZ4787*(::rrnB-3), λ⁻, *rph-1*, Δ(*rhaD-rhaB*)568,Δ*dgkA737::kan,hsdR514*) referred to as Δ*dgkA*, was used for screening parent inactive mutants and suppressors. Growth-defective parent inactive mutants (i.e. capable of growth only in media with high osmolarity) were screened by replica plating on selective media (*Wen et al., 1996*) (1% tryptone, 0.5% yeast extract, 0.01% arabinose, 1.6% agar, 34 μg/ml chloramphenicol and 50 μg/ml kanamycin) at decreasing concentrations of NaCl (0.15%, 0.03%, 0.01%, 0.007% and 0%), at 37°C for 12–16 hr. Identities of the parent inactive mutants were confirmed by Sanger sequencing. Each parent inactive mutant was used as template to randomize corresponding potential contact partners using the inverse PCR method discussed above, to construct X-ray and NMR structure specific second-site suppressor mutagenesis libraries. Screening of the libraries was carried out on selective media at varying concentrations of NaCl (mentioned above). Probable suppressors were identified as those which appear before the appearance of the corresponding parent inactive mutant on a plate at low NaCl concentration (<0.15%), that is restore the growth defect of the parent inactive mutant. WT DgkA and the empty vector pBAD-33 were used as positive and negative controls, respectively. Second-site putative suppressors were identified by Sanger sequencing at Macrogen, Korea. Fresh transformation of individual suppressors into *E. coli ΔdgkA* strain and comparison of the phenotype with the corresponding parent inactive mutant confirmed the result from the initial screen.

## Co-variation analysis for DgkA

Alignments of homologous sequences of DgkA were constructed by the jackhammer package of HMMER 3 (http://hmmer.janelia.org/search/jackhmmer) (*Finn et al., 2011*) with the number of iterations set to 3, E-value cut-off 1e-6, with a search against the nr database (http://www.ncbi.nlm.nih.gov/). Duplicate rows and gaps in the target sequence were removed from the multiple sequence alignment to yield a final set of 4175 alignments. This protocol has been adopted from (*Nugent and Jones, 2012*). The processed multiple sequence alignment was entered as input to DCA (*Morcos et al., 2011*), PSICOV (*Jones et al., 2012*) and SCA (*Halabi et al., 2009*) with default parameters as mentioned in the respective software packages. Web servers for GREMLIN (http://gremlin.bakerlab.org) (*Kamisetty et al., 2013*) and EVfold (http://evfold.org/evfold-web/evfold.do) (*Marks et al., 2011*) were used for analysis. The co-variation scores estimated by the above programs were used for analysis.

# Acknowledgements

We thank Prem Prakash Khushwaha for his help in cloning L36A/M63L CcdB and Chetana Baliga, Nandini Mani, Anushya P. and Farha Khan as well as other members of the RV laboratory for useful suggestions.

# Additional information

## Competing interests

AS, SK, PCJ, RV are authors on a patent application filed on behalf of the Indian Institute of Science, involving saturation suppressor mutagenesis methodology. The other author declares that no competing interests exist.

## Funding

| Funder | Grant reference number | Author |
|---|---|---|
| Department of Biotechnology , Ministry of Science and Technology | NO.BT/COE/34/SP15219/ 2015, DT.20/11/2015 | Raghavan Varadarajan |
| Department of Science and Technology, Ministry of Science and Technology | FNo.SB/SO/BB-0099/2013 DTD 24.6.14 | Raghavan Varadarajan |
| Council of Scientific and Industrial Research | Graduate Student Fellowship | Anusmita Sahoo Pankaj Jain |
| Department of Biotechnology , Ministry of Science and Technology | Graduate Student Fellowship | Shruti Khare |

The funders had no role in study design, data collection and interpretation, or the decision to submit the work for publication.

## Author contributions

AS, designed and performed the experiments, analysed the results, performed the computations and wrote the manuscript; SK, performed the computations and wrote the manuscript; SD, performed the experiments; PCJ, RV, designed the experiments, analysed the results and wrote the manuscript

# Additional files

## Supplementary files

• Supplementary File 1. Summary of sort details for different libraries of CcdB

• Supplementary File 2. Relative expression, binding and stabilities of parent inactive mutants and their suppressors in the case of CcdB

• Supplementary File 3. Sidechain-sidechain centroid distances and shortest distances between the listed residue pairs for putative differential contacts between DgkA X-ray and NMR structures. Experimentally observed (Parent inactive mutant, suppressor) pairs are indicated in bold. Main chain atoms and hydrogen atoms are not considered in the calculations.

## Major datasets

The following datasets were generated:

| Author(s) | Year | Dataset title | Dataset URL | Database, license, and accessibility information |
|---|---|---|---|---|
| Sahoo A, Khare S, Devanarayanan S, Jain P, Varadarajan R | 2015 | Data from: Residue proximity information and protein model discrimination using saturation-suppressor mutagenesis | http://dx.doi.org/10.5061/dryad.3g092 | Available at Dryad Digital Repository under a CC0 Public Domain Dedication |

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
