## [Decision Letter]

Thank you for submitting your work entitled "Residue proximity information and protein model discrimination using saturation-suppressor mutagenesis" for peer review at *eLife*. Your submission has been favorably evaluated by John Kuriyan (Senior editor) and three reviewers. One of three reviewers has agreed to reveal his identity: Dan Tawfik.

The reviewers have discussed the reviews with one another and the Reviewing editor has drafted this decision to help you prepare a revised submission.

Review:

The authors present a novel approach to determine 3D contacts in proteins using mutagenesis scans. They demonstrate the value of the inferred 3D contacts by showing their utility in discriminating between inaccurate and accurate 3D models. The experimental methodology involves an initial single mutagenesis scan that is used to identify residue mutations for a secondary mutation scan on that particular background.

In the case of the bacterial toxin CcdB, the single saturation mutagenesis scan was previously published by themselves (Adkar 2012). In this current work, they choose 5 deleterious mutations, based on a "RankScore" identified in their previous work. On each of the 5 backgrounds they did an exhaustive second site scan to find suppressors; i.e. those mutations that rescue the deleterious effect of the single mutation. They find a total of 10 suppressor mutations and then divide these into those likely to be contacts or not (proximal or distal) based on the RankScore of the suppressor mutation in the single saturation scan. Those secondary mutations with low deleteriousness – low RankScore – in the first scan are assumed to be 3D contacts.

The idea of using systematic suppressor-mutagenesis for the determination of protein structures is highly appealing, and constantly in 'the air'. Several groups are engaged in such efforts, and this paper may well be the first actual demonstration of what is likely to become a very powerful methodology. Specifically, the prediction demands a large number of sequences, and in such families, crystal structures are more likely to be available anyway. Membrane proteins comprise a particular challenge that this paper addresses.

Overall, this work is highly valuable, and the mechanisms of suppression are interesting. The manuscript, however, demands some rewriting, and a few critical points need to be addressed or clarified in a satisfactory manner before the paper can be considered acceptable for *eLife*.

Major concerns that must be addressed:

1) A key idea underlying the method is the discrimination between proximal and distal suppressor mutants, where the proximal ones will indicate physical contacts. The key result is given in Table 1. Based on this Table, 5 positions are mutated and 10 suppressor mutations tested. Thus, the crux of these methods (systematic suppressor mapping) is how to distinguish between distal and proximal suppressors, or global and local suppressors as they are often dubbed. The reviewers are not convinced that the selection criteria applied are objective and generic, namely applicable to any protein. The authors give no justification for the thresholds chosen, and do not show how the results would differ upon changes in these thresholds. The data in Table 1 suggest a very clear cutoff (the distant ones having RankScore 1, the contacting ones having ⩾ 26 scores) but the sample of sequences is so small, and this clearcut separation looks too good to be true.

Also, how do the authors exclude the active site? By checking the structure or blind? If blind the authors need to explain how.

To summarize, the reviewers are concerned that the classification of the residues in distal or proximal positions requires knowledge of the structure, or at least of the surface exposure. This is a limitation that may not make the method useful for structure prediction (or at least less useful). The revised paper must have a clear justification for the steps taken, explained cogently in the Introduction, and also later as necessary.

2) The application of the method to discriminate between different conformations of the DgkA multimers is an interesting one. The authors identify differential contacts between the 2 published conformations and present in Table 2 and Figure 6. However, some of the contacts identified as unique to one structure are really not at all far away in the other structure – even though closer in the example given. For instance, the M66-A99 minimum atom distance is 6.7 Å and 5.5 Å in the crystal structure 3ZE5 (inter monomers as with the NMR structures) and I67-E34 distance is 5.5 Å in the crystal structure 3ZE5 within the monomers (inter monomers in NMR structures) – therefore neither of these contacts is so unique to the NMR structures despite the domain swap, despite the fact they are closer in the crystal. Similarly, though much closer in the crystal, the min atom distance between V62- A41 (between monomers) in one of the NMR structures is 6.3 Å. Therefore, despite the domain swap, many 3D contacts are common to both structures within a looser definition, and therefore it is not clear that one can be sure that the structures are actually discriminated. It would be useful to tabulate the unique contacts with the minimum atom distance in both the structures including the multimeric assemblies.

3) The reviewers feel that global suppressors dominate compensation of highly destabilizing mutations, as applied here (e.g. see PMID: 17122770; PMID: 18495157). So the cases explored here, and especially dgkA, whereby all suppressors were local, may not reflect the difficulty of separating local/global in other proteins (e.g. see PMID: 25455030). Thus, the approach taken here need to be explained and justified with more rigor.

4) Comparison with other methods: The discussion of computational approaches to predict spatially proximal residues seems somewhat biased. It is not clear why EVFOLD, which is now deemed as the most successful method, was not examined. Further, one would like to see a comparison of the structural models offered by EVFOLD and other methods to both the X-ray and NMR structure. Some of these methods will give predictions that unambiguously coincide with the X-ray structure.

The result of the first test, i.e. selection of structures from a fold library may be less convincing than what the authors argue. The field of protein structure prediction went very deep into the use of decoy libraries to test prediction methods in the late 90's and early 2000's. It was later clear that discriminating structures in those conditions was a quite easy exercise and almost any simple method (i.e. surface exposition prediction) was able to perform well in that type of test. In other words, to be convincing these experiments will have to show that the discrimination is substantially better than the one provided by simple prediction methods.

To summarize the reviewers’ concerns, if the authors decide to include comparisons to other methods, then the comparisons should be done on fair basis. Two examples may be sufficient to illustrate the point. The prediction methods based on multiple sequence alignments are not specific for any of the proteins in the alignments and therefore it does not make sense to evaluate them in terms of distances between sidechains (that are specific to each structure), as the authors do. Second, and very important, since the publications of those prediction methods it was very clear that they can be applied *only* to rich alignments with thousands of sequences. The current application to small alignments does not demonstrate anything but a known principal limitation of the methods ("350 alignments" as reported in the paper).

The authors should decide whether or not to present comparisons to other methods in this paper, and present a judicious discussion of the value of their method.

Other issues to address:

1) While the rationale of detection of proximal suppressors is clear when it comes to protein stability, the effect on function does not follow this rationale. If you disturb a give side-chain-DNA contact, why would a mutation in an adjacent position compensate for that? Unlike the packing-stability that is accounted for (Figure 4), the mechanistic basis of compensation of functional residues remains therefore unclear. Perhaps the authors should not use "function", i.e. ligand binding, to illustrate the method (Figure 1) but rather, core-packing compensation.

2) Introduction, first sentence – “X-Ray crystallography and NMR…”. Cryo-EM is threatening to shadow both.

3) The Introduction/Discussion overlooks a large body of work that is highly relevant to this one. What the authors dub as systematic suppressor-mutagenesis was done before, often with the aim of unraveling epistatic interactions in proteins in relation to one, or few chosen positions (e.g. PMID: 20975933; 23935519), and sometimes in a systematic manner and in a context that is very similar to the authors (foremost, but not exclusively, PMID: 25455030). We're all prone to cite 'classics' (“Hecht and Sauer, 1985; Machingo et al, 2001; Pakula and Sauer, 1989; Sideraki et al, 2001”) and avoid recent literature that may compromise "novelty" but this does injustice to a lot of very good work, including, eventually, our own. The authors should do a more thorough search and reading, to provide an update picture of experimental explorations of covariance, or epistasis, in individual proteins, and specifically, in support of the claim that: "Though a small number of compensatory mutations were identified, in some cases these were ascertained to be spatially proximal while in others they were distal from the site of the original inactive mutation."

4) "In contrast to proximal suppressors, distal suppressors will typically be on the surface of protein and hence the individual suppressor mutation is expected to show WT like activity." This explanation is very confusing, also because the authors do not use "fitness" terms that would be easier to comprehend. What they mean in effect is that the suppressor mutation is expected to be neutral on its own, and beneficial, or compensatory in combination with the deleterious mutation (PIM in their terminology) – this would be positive sign epistasis in evolutionary terms. The meaning of the next sentence is completely unclear: "Further, unlike proximal suppressors no complementarity relative to the PIM is expected for a distal suppressor."

5) RankScore: 'residue depth' – define please, and also 'mutational tolerance'.

6) Subsection “Application of suppressor methodology to identify the functional conformation of the membrane protein DgkA in-vivo“: V62Q, M66S, M66L, I67V, V68G and W112V were identified as PIMs from screening of SSM – is this from previous work, or this one? If the latter, data need to be provided, and also the criteria for selecting these mutations for the suppressors screen.

7) The Abstract is somewhat misleading as one gets the impression that all possible compensatory pairs in the target proteins were identified. Upon reading further, it becomes clear that half a dozen deleterious mutations were chosen, and suppressors were identified for this small set. This does not make this work less valuable, on the contrary, it demonstrates its power to obtain structural information by exploring a relatively small number of positions. But it would be best to clarify this point.

8) One 'trick' of identifying global suppressors is that they in most cases comprise 'consensus/ancestral' mutations (see PMID: 18495157). The authors may wish to consider this as another parameter in their algorithm.

[Editors' note: further revisions were requested prior to acceptance, as described below.]

Thank you for resubmitting your work entitled "Residue proximity information and protein model discrimination using saturation-suppressor mutagenesis" for further consideration at *eLife*. Your revised article has been favorably evaluated by John Kuriyan (Senior editor) and three reviewers. The manuscript has been improved but there are some remaining issues that need to be addressed before acceptance, as outlined below:

Your manuscript has been read and discussed by three reviewers and the editor, and the decision is the work is, in principle, acceptable for publication in *eLife*. There are, however, a number of issues with the manuscript, as written, that can be improved by judicious editing. These issues are outlined below, and we ask that they be addressed in a revised manuscript. The revised manuscript will be handled by the editor, without further external review.

1) One of the reviewers is still not convinced that the method you use to distinguish between proximal/distal pairs, and to identify active site residues, is robust. You have provided a manuscript that has been submitted elsewhere that addresses the underlying concepts. Please provide a more complete justification for these points in the present manuscript, so that a non-specialist reader can fully understand the logic behind the method. Cite the submitted manuscript as appropriate. Refer to standard methods of analysis where appropriate, rather than to methods developed in your lab.

2) This reviewer is also not convinced that the superiority of this method is clearly demonstrated by the limited set of decoy calculations done and by the one EV-fold comparison that is provided. On the whole, though, we feel that these comparisons are useful, but you could make it more clear in the manuscript that these are illustrative differences rather than definitive demonstrations of superiority. Please point out limitations of the comparisons.

3) The manuscript is quite difficult for a non-specialist to follow, and in this way it obscures the innovation and depth of the work. It is essential that the manuscript be edited so that it is accessible to a generally knowledgeable structural biologist or biochemist. Some of the points to note are listed below, but it is recommended that the revised manuscript be read by a seasoned non-specialist colleague, and their advice taken, before being resubmitted to *eLife*.

A) The manuscript is heavy on long run-on paragraphs that introduce multiple ideas, making it difficult for the reader to follow. Break up the flow into separate paragraphs for each idea, concept or result.

B) Ideas are introduced out of order, often with highly specific and cryptic information provided before a more general explanation. For example, CcdB is used without explanation first. Later, concerning CcdB, the incomprehensible statement (to a non-specialist) that "This is <5L" is made. Only later is the biological function explained, and the "<5L" statement is never explained. This is but one of several such instances that make the paper difficult for a general audience.

C) Please avoid the use of inessential abbreviations – we are not under page limits in an online publication. For example, it is highly recommended that "parent inactivating mutation" be spelt out everywhere – the editor sees no reason why this should not be done, and the use of PIM is felt to add to the incomprehensibility of the manuscript. Likewise, why use SSM when it can be spelt out? Why abbreviate yeast surface display with the incomprehensible YSD? The editor asks that you retain only standard abbreviations or gene names.

D) Please ensure that all metrics are clearly explained to the non-specialist reader. For example, in the subsection “Discrimination between proximal and distal suppressors “"RankScore" is used with no explanation.

---

## [Author Response]

[…] Overall, this work is highly valuable, and the mechanisms of suppression are interesting. The manuscript, however, demands some rewriting, and a few critical points need to be addressed or clarified in a satisfactory manner before the paper can be considered acceptable for eLife.

*Major concerns that must be addressed:1) A key idea underlying the method is the discrimination between proximal and distal suppressor mutants, where the proximal ones will indicate physical contacts. The key result is given in Table 1. Based on this Table, 5 positions are mutated and 10 suppressor mutations tested. Thus, the crux of these methods (systematic suppressor mapping) is how to distinguish between distal and proximal suppressors, or global and local suppressors as they are often dubbed. The reviewers are not convinced that the selection criteria applied are objective and generic, namely applicable to any protein. The authors give no justification for the thresholds chosen, and do not show how the results would differ upon changes in these thresholds. The data in Table 1 suggest a very clear cutoff (the distant ones having RankScore 1, the contacting ones having* ⩾ *26 scores) but the sample of sequences is so small, and this clearcut separation looks too good to be true.*

Also, how do the authors exclude the active site? By checking the structure or blind? If blind the authors need to explain how.To summarize, the reviewers are concerned that the classification of the residues in distal or proximal positions requires knowledge of the structure, or at least of the surface exposure. This is a limitation that may not make the method useful for structure prediction (or at least less useful). The revised paper must have a clear justification for the steps taken, explained cogently in the Introduction, and also later as necessary.

1.1) Identification of active-site residues

Both buried and active-site residue positions possess high RankScores and high average mutational sensitivity (MS_seq_) values. Active-site residues can be distinguished from buried ones based on the pattern of mutational sensitivity (unpublished data). Mutational sensitivity (MS_seq_) was determined from the sequencing analysis of the single-site saturation library of CcdB (methodology described in (Adkar et al, 2012)). At buried positions (unpublished data), typically most aliphatic substitutions are tolerated. Polar and charged residues are poorly tolerated at buried positions. In contrast, mutations to aliphatic residues are often poorly tolerated at active-site residues (which are typically exposed). Polar and charged residues are sometimes tolerated and also the average mutational tolerance for active-site residues is typically lower than that for buried residues. Mutational tolerance is the fraction of active mutants for each mutant amino acid. This sensitivity to aliphatic substitutions at active-site residues is likely the reason that Ala scanning mutagenesis has been so successful at identifying residues at protein:protein and protein:ligand binding sites (Cunningham and Wells, 1989). Based on the above criteria residues Q2, F3, Y6, S22, I24, N95, W99, G100 and I101 can be identified as putative active-site residues based solely on the mutational data. Similar mutational patterns are seen for two other proteins for which extensive mutational data exist, the PDZ domain (PSD95^pdz3^) (McLaughlin et al, 2012) and the IgG-binding domain of protein G (GB1) (Olson et al., 2014) (unpublished data). In addition to differences in mutational sensitivity patterns, an important difference between active-site and buried-site mutations is that the former typically affect specific activity and not the level of properly folded protein, while the latter primarily affect the level of properly folded protein (Bajaj et al., 2008). Thus measurements of protein levels and possibly sensitivity of mutant activity to chaperone overexpression (Tokuriki and Tawfik, 2009) can also be used to distinguish between active-site and buried-site mutants. The average hydrophobicity and hydrophobic moment (Varadarajan et al., 1996) can also be used as supplementary parameters to distinguish between exposed, active-site and buried-site residues. The above points are included in the subsection "Selection of parent inactive mutant".

1.2) Classification of the suppressors as distal or proximal

In contrast to proximal suppressors, distal suppressors will typically be on the surface of protein (Bank et al., 2015) and hence the individual suppressor mutation is expected to be neutral in the WT background. Further, unlike proximal suppressors no complementarity in amino acid property relative to the parent inactive mutant (PIM) is expected for a distal suppressor. We have shown that the mutational sensitivity measure, RankScore correlates with residue depths derived from the crystal structure of the protein (Figure 4B Figure 3 of (Adkar et al., 2012)). Residue depth is defined as the distance of any atom/residue to the closest bulk water (Chakravarty and Varadarajan, 1999; Tan et al., 2011). The value for RankScore ranges between 1 to 100. For a given non active-site residue, a higher value of RankScore indicates that the residue is likely to be buried in the protein structure. The plot for variation of RankScore with residue depth is shown in Figure 9. Active-site residues (inferred solely from mutational data) and Gly and Pro residues have been removed. Also, only those residue positions for which data for ≥12 mutants is available are considered.

Author response image 1.Plot for variation of RankScore with respect to the residue depth.An average depth of ≥ 5.5Å corresponds to an average accessibility of ≤5% (Tan et al., 2014). The line parallel to X-axis indicates RankScore cutoff of 25 and the line parallel to Y-axis indicates depth of 5.5Å.**DOI:**
http://dx.doi.org/10.7554/eLife.09532.029

All the PIMs were chosen such that they are non-active-site and have a high RankScore. Hence, they are likely to be buried in the protein structure and in fact, are buried. It is therefore expected that positions at which local suppressors occur will also have high depth, high RankScores and high average mutational sensitivity in the SSM library (Figure 1). From Figure 9, it is evident that all the positions with low RankScores are exposed on the protein structure. Therefore, all positions with RankScore=1 are very likely to be present on the surface, and hence were classified as distal suppressors. A RankScore of 1 is a conservative cutoff for distal suppressors, probably a cutoff of five or ten would yield similar results. As can be seen from the figure, the RankScore cutoff of 25 which we have chosen for proximal suppressors, clearly identifies only buried residues with depth > 5.5Å. All these residues have accessibility < 1.5%. In the present work we assume that distal suppressors will be global suppressors. Hence, the most reliable way to identify these would be to confirm that the same putative distal suppressor is able to suppress multiple PIMs, preferably PIMs which are not in contact with each other. The latter is likely to be true if neither PIM has a suppressor at the site of the other PIM. Using these criteria we can clearly infer that R10G, E11R, E11K and E11P are likely global suppressors.

The section in the manuscript on ‘Discrimination between proximal and distal suppressors’ has been modified to include the additional information.

*2) The application of the method to discriminate between different conformations of the DgkA multimers is an interesting one. The authors identify differential contacts between the 2 published conformations and present in Table 2 and Figure 6. However, some of the contacts identified as unique to one structure are really not at all far away in the other structure* – *even though closer in the example given. For instance, the M66-A99 minimum atom distance is 6.7 Å and 5.5 Å in the crystal structure 3ZE5 (inter monomers as with the NMR structures) and I67-E34 distance is 5.5 Å in the crystal structure 3ZE5 within the monomers (inter monomers in NMR structures)* – *therefore neither of these contacts is so unique to the NMR structures despite the domain swap, despite the fact they are closer in the crystal. Similarly, though much closer in the crystal, the min atom distance between V62- A41 (between monomers) in one of the NMR structures is 6.3 Å. Therefore, despite the domain swap, many 3D contacts are common to both structures within a looser definition, and therefore it is not clear that one can be sure that the structures are actually discriminated. It would be useful to tabulate the unique contacts with the minimum atom distance in both the structures including the multimeric assemblies.*

It is unclear at this point, what the optimum cutoff distance should be for residue contact identification through suppressor mutagenesis. In the present study, true contacts were identified using a sidechain-sidechain centroid distance ≤7Å as the cutoff, for both X-ray and NMR structures of DgkA (as mentioned in the subsection “ContactScore as a model discriminator”). The NMR structure (2KDC) has co-ordinates for 16 poses. For the figures and distances in the paper we used only the first pose. However, we calculated the Root Mean Square Fluctuations (RMSF) for each residue (Cα atom) taking the average structure as the reference structure. Similar calculations were also performed with the first NMR pose as the reference structure. Both the plots are very similar. Figure 10 shows the distribution of the RMSF values for all residue positions when the first NMR pose is taken as the reference structure.

Author response image 2.Variation of RMSF w.r.t. residue positions.**DOI:**
http://dx.doi.org/10.7554/eLife.09532.030

Greater fluctuations are observed near the N-terminus of the protein (residues 1 to 33). Hence, we did not consider differential contacts within this region. Since, the region beyond the first 30 residues shows low RMS fluctuations, in Table 2 of the manuscript we report sidechain-sidechain centroid distances for residues showing differential contacts with the parent inactive mutant (PIM) in X-ray and the NMR 1^st^ pose structure. This discussion has been included in the subsection “Identification of differential contacts”. In [Supplementary-material SD3-data], we consider all NMR poses and report the closest distance between each PIM and all residues shortlisted in Table 2. The distances between sidechain centroids and the shortest distances between the two residues are shown. Main chain atoms are not considered. Since co-ordinates of hydrogen atoms are absent from the X-ray crystal structure, hydrogen atoms in the NMR structures are also not considered in the distance calculations.

With regard to the specific contacts mentioned by the reviewers, please see below.

A) 62 – 41 (X-ray contact): 6.3Å (NMR pose 4 – distance between VAL62 HG23 and ALA41 HB2 is 6.4Å.)

The above distances represent shortest distances whereas we consider centroid-centroid distances. Further, as hydrogen co-ordinates are absent from the crystal structure, we have not considered hydrogen co-ordinates in the NMR structures for distance calculations.

B) 66 – 99 (NMR contact): 6.7Å (X-ray – distance between MET66 CE and ALA99 O is 6.72 Å.)

The distance for this NMR contact is <7Å in the X-ray structure only if we consider the main chain atoms (which were not considered in the calculations). The side chains of these residues point away from each other in the X-ray structure.

C) 67 – 34 (NMR contact): 5.5Å (X-ray – distance between ILE67 N and GLU34 OE1 is 5.5Å.)

The distance for this NMR contact is <7Å in the X-ray structure only if we consider the main chain atoms (which were not considered in the calculations). The side chains of these residues point away from each other in the X-ray structure.

The shortlisted contacts in Table 2 contain a mixture of X-ray and NMR contacts. From all the shortlisted X-ray and NMR contacts, only a few suppressors could be experimentally identified. All these suppressors were consistent only with the contacts found in the crystal structure and are very far apart in the NMR structure.

3) The reviewers feel that global suppressors dominate compensation of highly destabilizing mutations, as applied here (e.g. see PMID: 17122770; PMID: 18495157). So the cases explored here, and especially dgkA, whereby all suppressors were local, may not reflect the difficulty of separating local/global in other proteins (e.g. see PMID: 25455030). Thus, the approach taken here need to be explained and justified with more rigor.

Section 1.2 above clarifies the basis for the cutoffs used to distinguish between local and global suppressors. We have shown that mutational sensitivity correlates well with residue depths for CcdB (Figure 4 of (Adkar et al., 2012)). We observed a similar high correlation for the PDZ domain (PSD95^pdz3^) (unpublished data). This high correlation of mutational sensitivity with depth would allow distinction of local and global suppressors for other globular proteins as well, as long as the majority of global suppressors lie on the surface. However, little is known about sensitivity to mutation in membrane proteins or natively unfolded proteins. Until such data becomes available we agree that it will be challenging to apply this methodology to these systems.

In the case of DgkA, we had the much simpler objective of distinguishing between two possible structures. While distinguishing between global and local suppressors maybe more challenging in membrane proteins, given sufficient double mutant data, it should be straightforward because global suppressors should suppress a much larger number of parent inactive mutants (PIMs) than local suppressors. We are currently confirming this by constructing larger double mutant libraries for both CcdB and DgkA. With regard to the relative frequency of global and local suppressors, if a single-site saturation mutagenesis library is enriched for inactive mutants (PIMs), subjected to random mutagenesis and screened for suppressors, the resulting population will be enriched for global suppressors (Bershtein et al., 2008; Bershtein et al., 2006). This is because a global suppressor will suppress multiple PIMs. As described in 1.2 above, this can be used to distinguish between local and global suppressors. However, for a specific PIM, it is not obvious that global suppressors will dominate. A recent study examined a library of the 75 amino acid RRM domain of the yeast poly-A binding protein (Melamed et al., 2013). Functional scores for 1246 single and 39,912 double mutants were obtained. Epistatic interactions were enriched for residue pairs with short sequence spacing (<5) and short distance (10-15Å). Another recent study (Olson et al., 2014) reported exhaustive screening of single and double mutants of GB1 (Olson et al, 2014). The majority of pairs displaying positive epistasis had Cβ-Cβ distances < 8Å. Both of the above studies indicate that local suppressors may occur at higher frequency than global ones with respect to individual PIMs but more data is required to confirm this. This discussion is added to the manuscript in the third paragraph of the Discussion section.

4) Comparison with other methods: The discussion of computational approaches to predict spatially proximal residues seems somewhat biased. It is not clear why EVFOLD, which is now deemed as the most successful method, was not examined. Further, one would like to see a comparison of the structural models offered by EVFOLD and other methods to both the X-ray and NMR structure. Some of these methods will give predictions that unambiguously coincide with the X-ray structure.

*The result of the first test, i.e. selection of structures from a fold library may be less convincing than what the authors argue. The field of protein structure prediction went very deep into the use of decoy libraries to test prediction methods in the late 90's and early 2000's. It was later clear that discriminating structures in those conditions was a quite easy exercise and almost any simple method (i.e. surface exposition prediction) was able to perform well in that type of test. In other words, to be convincing these experiments will have to show that the discrimination is substantially better than the one provided by simple prediction methods.To summarize the reviewers’ concerns, if the authors decide to include comparisons to other methods, then the comparisons should be done on fair basis. Two examples may be sufficient to illustrate the point. The prediction methods based on multiple sequence alignments are not specific for any of the proteins in the alignments and therefore it does not make sense to evaluate them in terms of distances between sidechains (that are specific to each structure), as the authors do. Second, and very important, since the publications of those prediction methods it was very clear that they can be applied only to rich alignments with thousands of sequences. The current application to small alignments does not demonstrate anything but a known principal limitation of the methods ("350 alignments" as reported in the paper).The authors should decide whether or not to present comparisons to other methods in this paper, and present a judicious discussion of the value of their method.*4.1) EVfold for contact prediction for DgkA

We thank the reviewers for the suggestion to use EVfold to generate structural models for DgkA. Residue coupling scores obtained by EVfold were plotted against the sidechain-sidechain centroid distances of residue pairs calculated from the X-ray and NMR structures. The plots are shown in Figure 11 respectively. Experimentally identified contacts from suppressor mutagenesis are shown in cyan.

Author response image 3.Residue coupling score obtained from EVfold is plotted against the sidechain-sidechain centroid distances calculated from a) the X-ray and b) the NMR structures.Blue lines parallel to the X-axis indicate the co-variation score of the L/2^th^ residue pair (when arranged in descending order of the score), where L is the length of the protein (L=121 for DgkA). Blue lines parallel to the Y-axis indicate a sidechain–sidechain centroid distance of 7Å which was used as the cutoff to define residue contacts. Experimentally determined spatially proximal (PIM, suppressor) contact pairs are shown in cyan.**DOI:**
http://dx.doi.org/10.7554/eLife.09532.031

The reviewers have pointed out that the co-variation prediction methods use a multiple sequence alignment (MSA) as input and the predictions therefore are not specific to the residues present at the predicted contact positions. Evaluating the performance of these methods based on sidechain-sidechain centroid distances is therefore not advisable. Hence, we plotted the co-variation scores obtained using EVfold against the Cα-Cα distances of residue pairs calculated from the X-ray and NMR structures. The plots are shown in Figure 12 respectively. Experimentally identified contacts from suppressor mutagenesis are shown in cyan.

Author response image 4.Residue coupling score obtained from EVfold is plotted against the Cα-Cα distances calculated from a) the X-ray and b) the NMR structures.Blue lines parallel to the X-axis indicate the co-variation score of the L/2^th^ residue pair (when arranged in descending order of the score), where L is the length of the protein (L=121 for DgkA). Blue lines parallel to the Y- axis indicate the distance of 7Å which was used as the cutoff to define residue contacts. Experimentally determined spatially proximal (PIM, suppressor) contact pairs are shown in cyan.**DOI:**
http://dx.doi.org/10.7554/eLife.09532.032

Several high scoring co-varying pairs predicted by EVfold were found to be proximal to each other (Figure 11, Figure 12) when mapped onto the crystal structure. However, there were other high scoring pairs which were either far apart in the X-ray structure or were in proximity when analyzed with the NMR structure. Overall, as with the other programs tested, the data from EVfold also are clearly more consistent with the X-ray structure. Of the six contacts identified from our suppressor analyses, three (62-41, 67-104 and 68-100) were predicted in the top L/2 co-varying pairs by EVfold (as with GREMLIN and PSICOV).

The distribution of predicted contact pairs appears better when sidechain-sidechain centroid distances are used (Figure 11) rather than when Cα-Cα distances are used (Figure 12) as more number of high scoring pairs are proximal to each other when the sidechain-sidechain centroid distance is computed. The results for prediction of co-variation using EVfold have been included in the manuscript in Figure 7 and Figure 7—figure supplement 1 and in the subsection “Computational approaches to predict spatially proximal residues”.

Prediction of 3D structural models is an additional feature of EVfold, not available in the other 4 methods tested (DCA, GREMLIN, SCA and PSICOV). We analyzed all the 50 models generated by EVfold. EVfold predicts only monomeric models. Hence, monomers of the X-ray and NMR structures were used for comparison. All the models were superimposed onto both X-ray and NMR structures using GROMACS. Backbone RMSDs were calculated.

All the EVfold models have a high RMSD in the range of 13 – 23 Å with both crystal and NMR structures. The models closest to crystal and NMR structures are shown in Figure 13 respectively. Thus while the EVfold co-variation scores are more consistent with the X-ray structure, the actual 3D structures predicted by EVfold are quite different from both X-ray and NMR structures, possibly because of the complication of modeling a homotrimeric, as opposed to a monomeric structure.

Author response image 5.Superimposition of EVfold model onto the (a) X-ray structure (PDB id: 3ZE5) and (b) the NMR structure of DgkA (PDB id: 2KDC).EVfold model is shown in tan and the X-ray and NMR structures are shown in light blue.**DOI:**
http://dx.doi.org/10.7554/eLife.09532.033

In accordance with the reviewers’ suggestion we have removed Figure 8 which shows the co-variation scores for CcdB obtained from a relatively small number of sequences. Our intention is not to downplay the very valuable information obtained from computational co-variation analysis, but rather to highlight the complementarity with experimental suppressor phenotypes. We have retained the co-variation data for DkgA (Figure 7) because this was obtained from a larger dataset of 4175 non-redundant homologous sequences. The computational co-variation predictions for DgkA, like our suppressor data, are clearly more consistent with the X-ray rather than the NMR structure of DgkA.

4.2) Comparison of RankScore and ContactScore with other simple methods for decoy discrimination

As suggested by the reviewers, we compared the decoy discrimination efficiency of rdepthscoreand ContactScore with another method which uses a simple scoring function based on residue accessiblity in globular proteins (Bahadur and Chakrabarti, 2009). The function (R_s_) evaluates the deviation from the average packing properties of all residues in a polypeptide chain corresponding to a model of its three-dimensional structure. R_s_ is calculated using the following formula, where ASA_xi_ and <ASA_x_> are the accessible surface area values of residue X at position *i* and the average ASA of residue X in a large dataset, respectively (Bahadur and Chakrabarti, 2009).

Rs=∑i = 1whole chain|ASAxi−<ASAx>|<ASAx>The parameter R_s_ was calculated using the above formula for the CcdB decoy set. Since R_s_ estimates deviation from the average ASAs, the native structure should ideally possess the lowest value of R_s_. However, when the CcdB decoy set was sorted according to the R_s_ values, the native structure was ranked 934^th^ and the correlation between RMSD and R_s_ was seen to be only 0.3. These data demonstrate that both rdepthscoreand Contact score (CSc) parameters derived from mutational data perform better than simple solvent accessibility based correlations such as the one observed above. In other work (manuscript in preparation) we show that incorporation of saturation mutagenesis phenotypic data can significantly improve the accuracy of CASP predictions. The calculations performed using the R_s_ parameter have been included in the last paragraph of the subsection “ContactScore as a model discriminator”.

Other issues to address:

*1) While the rationale of detection of proximal suppressors is clear when it comes to protein stability, the effect on function does not follow this rationale. If you disturb a give side-chain-DNA contact, why would a mutation in an adjacent position compensate for that? Unlike the packing-stability that is accounted for (Figure 4), the mechanistic basis of compensation of functional residues remains therefore unclear. Perhaps the authors should not use "function", i.e. ligand binding, to illustrate the method (Figure 1) but rather, core-packing compensation.*We chose ‘function’ to illustrate the method in Figure 1 because our readout for core packing compensation is a functional readout, i.e., ligand binding for CcdB and cell survival for DgkA. None of our PIMs is at a functional site for precisely the reason mentioned by the reviewers. In the present work we exclude active-site residues when choosing PIMs. The PIMs are present at the protein core and disturb the packing at the core. The resulting decrease in activity is due to a decrease in the amount of properly folded, functional protein in the cell. In the example given by the reviewers, the PIM is at a functional ‘active-site’ residue at a protein:ligand interface. In this situation we would look for a suppressor mutation in the binding partner (DNA in this case) and not an intragenic suppressor. Figure 1 is an overly simplified schematic. To clarify how core residue mutations affect activity we have now included an additional supplementary figure ( Figure 1—figure supplement 2).

2) Introduction, first sentence – “X-Ray crystallography and NMR…”. Cryo-EM is threatening to shadow both.

We agree and have added an appropriate sentence to the first paragraph of the Introduction. This methodology will be useful for any approach to integrative structure determination, including Cryo-EM.

3) The Introduction/Discussion overlooks a large body of work that is highly relevant to this one. What the authors dub as systematic suppressor-mutagenesis was done before, often with the aim of unraveling epistatic interactions in proteins in relation to one, or few chosen positions (e.g. PMID: 20975933; 23935519), and sometimes in a systematic manner and in a context that is very similar to the authors (foremost, but not exclusively, PMID: 25455030). We're all prone to cite 'classics' (“Hecht and Sauer, 1985; Machingo et al, 2001; Pakula and Sauer, 1989; Sideraki et al, 2001”) and avoid recent literature that may compromise "novelty" but this does injustice to a lot of very good work, including, eventually, our own. The authors should do a more thorough search and reading, to provide an update picture of experimental explorations of covariance, or epistasis, in individual proteins, and specifically, in support of the claim that: "Though a small number of compensatory mutations were identified, in some cases these were ascertained to be spatially proximal while in others they were distal from the site of the original inactive mutation."

We apologize for the omissions and have included additional citations and discussion of other published work, including the references mentioned above. The additional references are cited in the Introduction section and in the Discussion section.

*4) "In contrast to proximal suppressors, distal suppressors will typically be on the surface of protein and hence the individual suppressor mutation is expected to show WT like activity." This explanation is very confusing, also because the authors do not use "fitness" terms that would be easier to comprehend. What they mean in effect is that the suppressor mutation is expected to be neutral on its own, and beneficial, or compensatory in combination with the deleterious mutation (PIM in their terminology)* – *this would be positive sign epistasis in evolutionary terms. The meaning of the next sentence is completely unclear: "Further, unlike proximal suppressors no complementarity relative to the PIM is expected for a distal suppressor."*

We have included a sentence along the lines suggested by the reviewers to the subsection “Discrimination between proximal and distal suppressors”. The next sentence has also been modified to “Further, unlike proximal suppressors no complementarity in amino acid property relative to the PIM is expected for a distal suppressor.”

*5) RankScore: 'residue depth'* – *define please, and also 'mutational tolerance'.*

These definitions for ‘residue depth’ and ‘mutational tolerance’ are added in the sixth paragraph of the Introduction and in the Materials and Methods section.

6) Subsection “Application of suppressor methodology to identify the functional conformation of the membrane protein DgkA in-vivo“: V62Q, M66S, M66L, I67V, V68G and W112V were identified as PIMs from screening of SSM – is this from previous work, or this one? If the latter, data need to be provided, and also the criteria for selecting these mutations for the suppressors screen.

The residues V62, M66, I67, V68 and W112 all have differential contacts in the X-ray and NMR structure and hence were chosen as sites for introducing PIMs. The details are described in the Methods section in the subsection “Identification of differential contacts”. Briefly, each of the above residues was individually randomized and the resulting small libraries were screened for inactive mutants. Charged and aromatic substitutions were excluded from the finally selected PIMs as we thought they would be highly destabilizing and it might not be possible to obtain suppressors. Phenotypic data for all selected PIMs and their suppressors are shown in Figure 6 and Figure 6—figure supplement 2. The only exception is the PIM W112V for which no suppressor was obtained.

7) The Abstract is somewhat misleading as one gets the impression that all possible compensatory pairs in the target proteins were identified. Upon reading further, it becomes clear that half a dozen deleterious mutations were chosen, and suppressors were identified for this small set. This does not make this work less valuable, on the contrary, it demonstrates its power to obtain structural information by exploring a relatively small number of positions. But it would be best to clarify this point.

The Abstract has been modified to clarify this point.

8) One 'trick' of identifying global suppressors is that they in most cases comprise 'consensus/ancestral' mutations (see PMID: 18495157). The authors may wish to consider this as another parameter in their algorithm.

We thank the reviewers for this suggestion and have mentioned this in the Discussion section. The CcdB consensus sequence was obtained from a multiple sequence alignment (MSA) of 350 homologs using MATLAB and the ancestral sequence was obtained using the FastML server (Ashkenazy et al., 2012). The likely global suppressors we obtained in the present study are R10G, E11R and E11K. At R10 the ancestral and consensus amino acids are P and R respectively and at E11 they are A and N respectively. Hence, at least for these two positions, the ancestral/consensus amino acids were different from the experimentally obtained suppressors. Further experiments are required to ascertain whether the ancestral/consensus amino acids will also act as global suppressors.

References:

Tan KP, Khare S, Varadarajan R, Madhusudhan MS (2014) TSpred: a web server for the rational design of temperature-sensitive mutants. Nucleic Acids Res 42: W277-284.

Cunningham BC, Wells JA (1989) High-resolution epitope mapping of hGH-receptor interactions by alanine-scanning mutagenesis. Science 244: 1081-1085.

[Editors' note: further revisions were requested prior to acceptance, as described below.]

Your manuscript has been read and discussed by three reviewers and the editor, and the decision is the work is, in principle, acceptable for publication in eLife. There are, however, a number of issues with the manuscript, as written, that can be improved by judicious editing. These issues are outlined below, and we ask that they be addressed in a revised manuscript. The revised manuscript will be handled by the editor, without further external review.

1) One of the reviewers is still not convinced that the method you use to distinguish between proximal/distal pairs, and to identify active site residues, is robust. You have provided a manuscript that has been submitted elsewhere that addresses the underlying concepts. Please provide a more complete justification for these points in the present manuscript, so that a non-specialist reader can fully understand the logic behind the method. Cite the submitted manuscript as appropriate. Refer to standard methods of analysis where appropriate, rather than to methods developed in your lab.

We have described the criteria used to distinguish between proximal and distal suppressors in the subsection “Discrimination between proximal and distal suppressors”. We have also included an additional Figure, Figure 1—figure supplement 1 with representative mutational data at active-site, buried and exposed, non-active site positions. The mutational criteria to distinguish between proximal and distal suppressors work for at least three proteins CcdB, PDZ domain (PSD95^pdz3^) and the IgG-binding domain of protein G (GB1). How well these will work for other proteins remains to be seen. In addition, since it is likely that distal suppressors will be global suppressors, they will suppress parent inactive mutants at multiple residue positions, including ones which are not in contact with each other. Hence putative distal suppressors can be introduced into the background of multiple parent inactive mutants to confirm that these are indeed global (and therefore distal) suppressors. This is mentioned in the aforementioned subsection. We have largely referred to standard methods of analysis wherever possible. The submitted manuscript has been cited as ‘unpublished work’ since we still do not have confirmation on when and where it will eventually be published.

2) This reviewer is also not convinced that the superiority of this method is clearly demonstrated by the limited set of decoy calculations done and by the one EV-fold comparison that is provided. On the whole, though, we feel that these comparisons are useful, but you could make it more clear in the manuscript that these are illustrative differences rather than definitive demonstrations of superiority. Please point out limitations of the comparisons.

We agree that this work represents an initial proof of principle and is by no means definitive. We view this approach as complementary, not superior to computational methods to identify correlated mutations and have stated this in the Introduction, Results and Discussion sections. We have discussed limitations of our experimental methodology in the first few paragraphs of the Discussion, stated that further tests to assess its general utility are required, and have suggested improvements to the methodology in the concluding paragraphs of the Discussion section.

3) The manuscript is quite difficult for a non-specialist to follow, and in this way it obscures the innovation and depth of the work. It is essential that the manuscript be edited so that it is accessible to a generally knowledgeable structural biologist or biochemist. Some of the points to note are listed below, but it is recommended that the revised manuscript be read by a seasoned non-specialist colleague, and their advice taken, before being resubmitted to eLife.

We have made extensive efforts to improve the clarity of the manuscript and have received useful inputs from four other colleagues unfamiliar with the work who were given the manuscript to read. All their suggestions, as well as those of the *eLife* reviewers, have been incorporated.

A) The manuscript is heavy on long run-on paragraphs that introduce multiple ideas, making it difficult for the reader to follow. Break up the flow into separate paragraphs for each idea, concept or result.

Done.

B) Ideas are introduced out of order, often with highly specific and cryptic information provided before a more general explanation. For example, CcdB is used without explanation first. Later, concerning CcdB, the incomprehensible statement (to a non-specialist) that "This is <5L" is made. Only later is the biological function explained, and the "<5L" statement is never explained. This is but one of several such instances that make the paper difficult for a general audience.

Changes have been appropriately made to introduce ideas in their proper context.

*C) Please avoid the use of inessential abbreviations* – *we are not under page limits in an online publication. For example, it is highly recommended that "parent inactivating mutation" be spelt out everywhere* – *the editor sees no reason why this should not be done, and the use of PIM is felt to add to the incomprehensibility of the manuscript. Likewise, why use SSM when it can be spelt out? Why abbreviate yeast surface display with the incomprehensible YSD? The editor asks that you retain only standard abbreviations or gene names.*

D) Please ensure that all metrics are clearly explained to the non-specialist reader. For example, in the subsection “Discrimination between proximal and distal suppressors “"RankScore" is used with no explanation.

The use of abbreviations has now been minimized except in the case of a few figures where space is limited. Even in those cases, the abbreviation is clearly explained in the figure caption. All metrics are explained and additional sections have been added to the start of the Methods section for this purpose.